# Continuous-Time Functional Diffusion Processes

**Giulio Franzese**
EURECOM, France

**Giulio Corallo**
EURECOM, France

**Simone Rossi**[*]
Stellantis, France

**Markus Heinonen**
Aalto University, Finland

**Maurizio Filippone**
EURECOM, France

**Pietro Michiardi**
EURECOM, France

## Abstract

We introduce Functional Diffusion Processes (FDPs), which generalize score-based diffusion models to infinite-dimensional function spaces. FDPs require a new mathematical framework to describe the forward and backward dynamics, and several extensions to derive practical training objectives. These include infinite-dimensional versions of Girsanov theorem, in order to be able to compute an ELBO, and of the sampling theorem, in order to guarantee that functional evaluations in a countable set of points are equivalent to infinite-dimensional functions. We use FDPs to build a new breed of generative models in function spaces, which do not require specialized network architectures, and that can work with any kind of continuous data. Our results on real data show that FDPs achieve high-quality image generation, using a simple MLP architecture with orders of magnitude fewer parameters than existing diffusion models. Code available here.

## 1 Introduction

Diffusion models have recently gained a lot of attention both from academia and industry. The seminal work on denoising diffusion (Sohl-Dickstein et al., 2015) has spurred interest in the understanding of such models from several perspectives, ranging from denoising autoencoders (Vincent, 2011) with multiple noise levels (Ho et al., 2020), variational interpretations (Kingma et al., 2021), annealed (Song & Ermon, 2019) and continuous-time score matching (Song & Ermon, 2020; Song et al., 2021). Several recent extensions of the theory underpinning diffusion models tackle alternatives to Gaussian noise (Bansal et al., 2022; Rissanen et al., 2022), second order dynamics (Dockhorn et al., 2022), and improved training and sampling (Xiao et al., 2022; Kim et al., 2022b; Franzese et al., 2022).

Diffusion models have rapidly become the go-to approach for generative modeling, surpassing GANs (Dhariwal & Nichol, 2021) for image generation, and have recently been applied to various modalities such as audio (Kong et al., 2021; Liu et al., 2022), video (Ho et al., 2022; He et al., 2022), molecular structures and general 3D shapes (Trippe et al., 2022; Hoogeboom et al., 2022; Luo & Hu, 2021; Zeng et al., 2022). Recently, the generation of diverse and realistic data modalities (images, videos, sound) from open-ended text prompts (Ramesh et al., 2022; Saharia et al., 2022; Rombach et al., 2022) has projected practitioners into a whole new paradigm for content creation.

A common trait of diffusion models is the need to understand their design space (Karras et al., 2022), and tailor the inner working parts to the chosen application and data domain. Diffusion models require specialization, ranging from architectural choices of neural networks used to approximate the score (Dhariwal & Nichol, 2021; Karras et al., 2022), to fine-grained details such as an appropriate definition of a noise schedule (Dhariwal & Nichol, 2021; Salimans & Ho, 2022), and mechanisms to deal with resolution and scale (Ho et al., 2021). Clearly, the data domain impacts profoundly such design choices. As a consequence, a growing body of work has focused on the projection of data

---

[*]This work was done while working at EURECOM

37th Conference on Neural Information Processing Systems (NeurIPS 2023).

modalities into a latent space (Rombach et al., 2022), either by using auxiliary models such as a VAEs (Vahdat et al., 2021), or by using a functional data representation (Dupont et al., 2022a). These approaches lead to increased efficiency, because they operate on smaller dimensional spaces, and constitute a step toward broadening the applicability of diffusion models to general data.

The idea of modelling data with continuous functions has several advantages (Dupont et al., 2022a): it allows working with data at arbitrary resolutions, it enjoys improved memory-efficiency, and it allows simple architectures to represent a variety of data modalities. However, a theoretically grounded understanding of how diffusion models can operate directly on continuous functions has been elusive so far. Preliminary studies apply established diffusion algorithms on a discretization of functional data by conditioning on point-wise values (Dutordoir et al., 2022; Zhuang et al., 2023). A line of work that is closely related to ours include approaches such as Kerrigan et al. (2022), who consider a Gaussian noise corruption process in Hilbert space and derive a loss function formulated on infinite-dimensional measures to approximate the conditional mean of the reverse process. Within this line of works, Mittal et al. (2022) consider diffusion of Gaussian processes. We are aware of other concurrent works that study diffusion process in Hilbert spaces (Lim et al., 2023; Pidstrigach et al., 2023; Hagemann et al., 2023). However, differently from us, these works do not formally prove that the score matching optimization is a proper evidence lower bound (ELBO), but simply propose it as an heuristic. None of these prior works discuss the limits of discretization, resulting in the failure of identifying which subset of functions can be reconstructed through sampling. Finally, the parametrization we present in our work merges how functions and score are approximated using a single, simple model.

The main goal of our work is to deepen our understanding of diffusion models in function space. We present a new mathematical framework to lift diffusion models from finite-dimensional inputs to function spaces, contributing to a general method for data represented by continuous functions.

In § 2, we present Functional Diffusion Processs (FDPs), which generalize diffusion processes to infinite-dimensional functional spaces. We define forward (§ 2.1) and backward (§ 2.2) FDPs, and consider *generic* functional perturbations, including noising and Laplacian blurring. Using an extension of Girsanov theorem, we derive in § 2.3 an ELBO, which allows defining a parametric model to approximate the score of the functional density of FDPs. Given a FDP and the associated ELBO, we are one-step closer to the definition of a loss function to learn the parametric score. However, our formulation still resides in an abstract, infinite-dimensional Hilbert space.

Then, for practical reasons, in § 3, we specify for which subclass of functions we can perfectly reconstruct the original function given only its evaluation in a countable set of points. This is an extension of the sampling theorem, which we use to move from the infinite-dimensional domain of functions to a finite-dimensional domain of discrete mappings.

In § 4, we discuss various options to implement such discrete mappings. In this work, we explore in particular the usage of implicit neural representations (INRs) (Sitzmann et al., 2020) and Transformers Vaswani et al. (2017) to jointly model both the sampled version of infinite-dimensional functions, and the score network, which is central to the training of FDPs, and is required to simulate the backward process. Our training procedure, discussed in § 5, involves approximate, finite-dimensional Stochastic Differential Equations (SDEs) for the forward and backward processes, as well as for the ELBO.

We complement our theory with a series of experiments to illustrate the viability of FDPs, in § 6. In our experiments, the score network is a simple multilayer perceptron (MLP), with several orders of magnitude fewer parameters than any existing score-based diffusion model. To the best of our knowledge, we are the first to show that a functional-space diffusion model can generate realistic image data, beyond simple data-sets and toy models.

## 2 Functional Diffusion Processes (FDPs)

We begin by defining diffusion processes in Hilbert Spaces, which we call functional diffusion processes (FDPs). While the study of diffusion processes in Hilbert spaces is not new (Föllmer & Wakolbinger, 1986; Millet et al., 1989; Da Prato & Zabczyk, 2014), our objectives depart from prior work, and call for an appropriate treatment of the intricacies of FDPs, when used in the context of generative modeling. In § 2.1 we introduce a generic class of diffusion processes in Hilbert spaces. The key object is Equation (1), together with its associated path measure $\mathbb{Q}$ and the time varying

measure $\rho_t$, where $\rho_0$ represents the starting (data) measure. In § 2.2 we derive the reverse FDP with the associated path-reversed measure $\hat{\mathbb{Q}}$, and in § 2.3 we use an extension of Girsanov theorem for infinite-dimensional SDEs to compute the ELBO. The ELBO is a training objective involving a generalization of the score function (Song et al., 2021).

## 2.1 The forward diffusion process

We consider $H$ to be a real, separable Hilbert space with inner product $\langle \cdot, \cdot \rangle$, norm $\|\cdot\|_H$, and countable orthonormal basis $\{e^k\}_{k=1}^{\infty}$. Let $L(H)$ be the set of bounded linear operators on $H$, $B(H)$ be its Borel $\sigma-$algebra, $B_b(H)$ be the set of bounded $B(H)-$measurable functions $H \to \mathbb{R}$, and $P(H)$ be the set of probability measures on $(H, B(H))$. Consider the following $H$-valued SDE:

$$\begin{cases} \mathrm{d}X_t = (\mathcal{A}X_t + f(X_t, t))\,\mathrm{d}t + \mathrm{d}W_t, \\ X_0 \sim \rho_0 \in P(H), \end{cases} \tag{1}$$

where $t \in [0, T]$, $W_t$ is a $R-$Wiener process on $H$ defined on the quadruplet $(\Omega, \mathcal{F}, (\mathcal{F}_t)_{t \geq 0}, \mathbb{Q})$, and $\Omega, \mathcal{F}$ are the sample space and canonical filtration, respectively. The domain of $f$ is $D(f) \in B([0, T] \times H)$, where $f : D(f) \subset [0, T] \times H \to H$ is a measurable map. The operator $\mathcal{A} : D(\mathcal{A}) \subset H \to H$ is the infinitesimal generator of a $C_0-$ semigroup $\exp(t\mathcal{A})$ in $H$ ($t \geq 0$), and $\rho_0$ is a probability measure in $H$. We consider $\Omega$ to be $C^1([0, T])$, that is the space of all continuous mappings $[0, T] \to H$, and $X_t(\omega) = \omega(t), \omega \in \Omega$ to be the *canonical* process. The requirements on the terms $\mathcal{A}, f$ that ensure existence of solutions to Equation (1) depend on the type of noise — *trace-class* ($\mathrm{Tr}\{R\} < \infty$) or *cylindrical* ($R = I$) — used in the FDP (Da Prato & Zabczyk (2014), *Hypothesis 7.1* or *Hypothesis 7.2* for trace-class and cylindrical noise, respectively).

The measure associated with Equation (1) is indicated with $\mathbb{Q}$. The *law* induced at time $\tau \in [0, T]$ by the canonical process on the measure $\mathbb{Q}$ is indicated with $\rho_\tau \in P(H)$, where $\rho_\tau(S) = \mathbb{Q}(\{\omega \in \Omega : X_\tau(\omega) \in S\})$, and $S$ is any element of $\mathcal{F}$. Notice, that in infinite dimensional spaces there is not an equivalent of the Lebesgue measure to get densities from measures. In our case we consider however, when it exists, the single dimensional density $\rho_\tau^{(d)}(x^i | x^{j \neq i})$, defined implicitly through $\mathrm{d}\rho_\tau(x^i | x^{j \neq i}) = \rho_\tau^{(d)}(x^i | x^{j \neq i})\mathrm{d}x^i$, being $\mathrm{d}x^i$ the Lebesgue measure. To avoid cluttering the notation, in this work we simply shorten $\rho_\tau^{(d)}(x^i | x^{j \neq i})$ with $\rho_\tau^{(d)}(x)$ whenever unambiguous. In Appendix B we provide additional details on the time-varying measure $\rho_t(\mathrm{d}x)\mathrm{d}t$. Before proceeding, it is useful to notice that Equation (1) can also be expressed as an (infinite) system of stochastic differential equations in terms of $X_t^k = \langle X_t, e^k \rangle$ as:

$$\mathrm{d}X_t^k = b^k(X_t, t)\mathrm{d}t + \mathrm{d}W_t^k, \quad k = 1, \ldots, \infty, \tag{2}$$

where we introduced the projection $b^k(X_t, t) = \langle \mathcal{A}X_t + f(X_t, t), e^k \rangle$. Moreover, $\mathrm{d}W_t^k = \langle dW_t, e^k \rangle$ with covariance given by $\mathbb{E}[W_t^k W_s^j] = \delta(k - j)r^k \min(s, t)$, $\delta$ in Kroenecker sense, and $r^k$ is the projection on the base of $R$.

## 2.2 The reverse diffusion process

We now derive the reverse time dynamics for FDPs of the form defined in Equation (1). We require that the time reversal of the canonical process, $\hat{X}_t = X_{T-t}$, is again a diffusion process, with distribution given by the **path-reversed** measure $\hat{\mathbb{Q}}(\omega)$, along with the reversed filtration $\hat{\mathcal{F}}$. Note that the time reversal of an infinite dimensional process is more involved than for the finite dimensional case (Anderson, 1982; Föllmer, 1985). There are two major approaches to guarantee the existence of the reverse diffusion process. The first approach (Föllmer & Wakolbinger, 1986) is applicable only when $R = I$ (the case of cylindrical Wiener processes) and it relies on a finite local entropy condition. The second approach, which is valid in the case of trace class noise $\mathrm{Tr}\{R\} < \infty$, is based on stochastic calculus of variations (Millet et al., 1989). The full technical analysis of the necessary assumptions for the two approaches is involved, and we postpone formal details to Appendix A.

**Theorem 1.** *Consider Equation* (1). *If $R = I$, suppose Assumption 1 in Appendix A.1 holds; else, ($R \neq I$) suppose Assumption 5 annd Assumption 6 in Appendix A.2 hold. Then $\hat{X}_t$, corresponding to the path measure $\hat{\mathbb{Q}}(\omega)$, has the following SDE representation:*

$$\begin{cases} d\hat{X}_t = \left( -\mathcal{A}\hat{X}_t - f(\hat{X}_t, T - t) + RD_x \log \rho_{T-t}^{(d)}(\hat{X}_t) \right) dt + d\hat{W}_t, \\ \hat{X}_0 \sim \rho_T, \end{cases} \tag{3}$$

where $\hat{W}$ is a $\hat{\mathbb{Q}}$ $R-$Wiener process, and the notation $D_x \log \rho_t^{(d)}(x)$ stands for the mapping $H \to H$ that, when projected, satisfies $\langle D_x \log \rho_t^{(d)}(x), e^k \rangle = \frac{\partial}{\partial x^k} \log \left( \rho_t^{(d)}(x^k \mid x^{i \neq k}) \right)$.

*By projecting onto the eigenbasis, we have an infinite system of* SDEs:

$$d\hat{X}_t^k = \left( -b^k(\hat{X}_t, T - t) + r^k \frac{\partial}{\partial x^k} \log \left( \rho_{T-t}^{(d)}(\hat{X}_t) \right) \right) dt + d\hat{W}_t^k, \quad k = 1, \ldots, \infty. \tag{4}$$

The methodology proposed in this work requires to operate on proper Wiener processes, with $\mathrm{Tr}\{R\} < \infty$, which implies, intuitively, that the considered noise has finite variance. We now discuss a Corollary, in which Assumption 5 is replaced by stricter conditions, that we use to check the validity of the practical implementation of FDPs.

**Corollary 1.** *Suppose Assumption 6 from Appendix A.2 holds. Assume that i)* $\mathrm{Tr}\{R\} = \sum_i r^i < \infty$, *ii)* $b^i(x, t) = b^i x^i, \forall i$, *i.e. the drift term is linear and only depends on $x$ through its projection onto the corresponding basis and iii) the drift is bounded, such that* $\exists K > 0 : -K < b^i < 0, \forall i$. *Then, the reverse process evolves according to Equation* (4).

Theorem 1 stipulates that, given some conditions, the reverse time dynamics for FDPs of the form defined in Equation (1) exist. Our analysis provides theoretical grounding to the observations in concurrent work (Lim et al., 2023) where, empirically, it is observed that the cylindrical class of noise is not suitable. We argue that, when $R = I$, the difficulty stems from designing the coefficients $b^i$ of the SDEs such that the forward (see requirement (5.3) in Da Prato & Zabczyk (2014)) as well as the backward processes Assumption 1 exist. The work by Bond-Taylor & Willcocks (2023) uses cylindrical (white) noise, but we are not aware of any theoretical justification, since the model architecture is only partially suited for the functional domain.

As an addendum, we note that the advantages of projecting the forward and backward processes on the eigenbasis of the Hilbert space $H$, as in Equation (2) and Equation (4), become evident when discussing about the implementation of FDPs, specifically when we derive practical expressions for training and the simulation of the backward process, as discussed in § 5, and in a fully expanded toy example in Appendix D.

### 2.3 A Girsanov formula for the ELBO

Direct simulation of the backward FDP described by Equation (3) is not possible. Indeed, we have no access to the **true score** of the density $\rho_\tau^{(d)}$ induced at time $\tau \in [0, T]$. To solve the problem, we introduce a **parametric score function** $s_{\boldsymbol{\theta}} : H \times [0, T] \times \mathbb{R}^m \to H$. We consider the dynamics:

$$\begin{cases} d\hat{X}_t = \left( -\mathcal{A}\hat{X}_t - f(\hat{X}_t, T - t) + Rs_{\boldsymbol{\theta}}(\hat{X}_t, T - t) \right) dt + d\tilde{W}_t, \\ \hat{X}_0 \sim \chi_T \in P(H), \end{cases} \tag{5}$$

with path measure $\hat{\mathbb{P}}^{\chi_T}$, and $d\tilde{W}_t$ being a $\hat{\mathbb{P}}^{\chi_T}$ $R-$Wiener process. To emphasize the connection between Equation (3) and Equation (5), we define initial conditions with the subscript $T$, instead of 0. In principle, we should have $\chi_T = \rho_T$, as it will be evident from the ELBO in Equation (8). However, $\rho_T$ has a simple and easy-to-sample-from distribution only for $T \to \infty$, which is not compatible with a realistic implementation. The analysis of the discrepancy when $T$ is finite is outside of the scope of this work, and the interested reader can refer to Franzese et al. (2022) for an analysis on standard diffusion models. The final measure of the new process at time $T$ is indicated by $\chi_0$, i.e. $\chi_0(S) = \hat{\mathbb{P}}^{\chi_T}(\{\omega \in \Omega : \hat{X}_T(\omega) \in S\})$.

Next, we quantify the discrepancy between $\chi_0$ and the true data measure $\rho_0$ through an ELBO. Thanks to an extension of Girsanov theorem to infinite dimensional SDEs (Da Prato & Zabczyk, 2014), it is possible to relate the path measures ($\hat{\mathbb{Q}}$ and $\hat{\mathbb{P}}^{\chi_T}$, respectively) of the process $\hat{X}_t$ induced by different drift terms in Equation (3) and different initial conditions.

Starting from the score function $s_{\boldsymbol{\theta}}$, we define:

$$\gamma_{\boldsymbol{\theta}}(x,t) = R\left(s_{\boldsymbol{\theta}}(x,T-t) - D_x \log \rho_{T-t}^{(d)}(x)\right). \tag{6}$$

Under loose regularity assumptions (see Condition 2 in Appendix A.4) $\tilde{W}_t = \hat{W}_t - \int_0^t \gamma_{\boldsymbol{\theta}}(X_s,s)ds$ is a $\hat{\mathbb{P}}^{\rho_T}$ $R$–Wiener process (Theorem 10.14 in Da Prato & Zabczyk (2014)), where Girsanov Theorem also tells us that the measure $\hat{\mathbb{P}}^{\rho_T}$ satisfies the Radon-Nikodym derivative:

$$\frac{d\hat{\mathbb{P}}^{\rho_T}}{d\hat{\mathbb{Q}}} = \exp\left(\int_0^T \langle \gamma_{\boldsymbol{\theta}}(\hat{X}_t,t), d\hat{W}_t \rangle_{R^{\frac{1}{2}}H} - \frac{1}{2}\int_0^T \left\|\gamma_{\boldsymbol{\theta}}(\hat{X}_t,t)\right\|_{R^{\frac{1}{2}}H}^2 dt\right). \tag{7}$$

By virtue of the disintegration theorem, $d\hat{\mathbb{Q}} = d\hat{\mathbb{Q}}_0 d\rho_T$ and similarly $d\hat{\mathbb{P}}^{\rho_T} = d\hat{\mathbb{P}}_0 d\rho_T$, being $\hat{\mathbb{Q}}_0, \hat{\mathbb{P}}_0$ the measures of the processes when considering a particular initial value. Then, $\hat{\mathbb{P}}^{\chi_T}$ satisfies $d\hat{\mathbb{P}}^{\chi_T} = d\hat{\mathbb{P}}\frac{d\chi_T}{d\rho_T}$, for any measure $\chi_T \in P(H)$. Consequently, the canonical process $\hat{X}_t$ has an SDE representation according to Equation (5), under the new path measure $\hat{\mathbb{P}}^{\chi_T}$. Then (see Appendix A.5 for the derivation) we obtain the ELBO:

$$\mathrm{KL}\left[\rho_0 \| \chi_0\right] \le \frac{1}{2}\mathbb{E}_{\mathbb{Q}}\left[\int_0^T \|\gamma_{\boldsymbol{\theta}}(X_t,t)\|_{R^{\frac{1}{2}}H}^2 dt\right] + \mathrm{KL}\left[\rho_T \| \chi_T\right]. \tag{8}$$

Provided that the required assumptions in Theorem 1 are met, the validity of Equation (8) is general. Our goal, however, is to set the stage for a practical implementation of FDPs, which calls for design choices that easily enable satisfying the required assumptions for the theory to hold. Then, for the remainder of the paper, we consider the particular case where $f = 0$ in Equation (1). This simplifies the dynamics as follows:

$$dX_t = \mathcal{A}X_t dt + dW_t, \quad X_0 \sim \rho_0 \in P(H) \tag{9}$$

$$d\hat{X}_t = \left(-\mathcal{A}\hat{X}_t + Rs_{\boldsymbol{\theta}}(\hat{X}_t, T-t)\right)dt + d\tilde{W}_t, \quad \hat{X}_0 \sim \chi_T \in P(H) \tag{10}$$

Since the only drift component in Equation (9) is the linear term $\mathcal{A}$, the projection $b^j$ will be linear as well. Such a design choice, although not necessary from a theoretical point of view, carries several advantages. The design of a drift term satisfying the conditions of Corollary 1 becomes straightforward, where such conditions naturally aligns with the requirements of the existence of the forward process (Chapter 5 of Da Prato & Zabczyk (2014)). Moreover, the forward process conditioned on given initial conditions admits known solutions, which means that simulation of SDE paths is cheap and straightforward, without the need for performing full numerical integration. Finally, it is possible to claim existence of the **true score function** and even provide its analytic expression (full derivation in Appendix A.7) as:

$$D_x \log \rho_t^{(d)}(x) = -\mathcal{S}(t)^{-1}\left(x - \exp(t\mathcal{A})\mathbb{E}\left[X_0 \mid X_t = x\right]\right), \tag{11}$$

where $\mathcal{S}(t) = \left(\int_{s=0}^t \exp((t-s)\mathcal{A})R\exp((t-s)\mathcal{A}^\dagger)ds\right)$. This last aspect is particularly useful when considering the conditional version of Equation (6), through $\langle D_x \log \rho_t^{(d)}(x \mid x_0), e^k \rangle = \frac{\partial}{\partial x^k}\log\left(\rho_t^{(d)}(x^k \mid x^{i \neq k}, x_0)\right)$, as:

$$\tilde{\gamma}_{\boldsymbol{\theta}}(x, x_0, t) = R\left(s_{\boldsymbol{\theta}}(x, T-t) - D_x \log \rho_{T-t}^{(d)}(x \mid x_0)\right), \tag{12}$$

where, similarly to the unconditional case, we have $D_x \log \rho_t^{(d)}(x \mid x_0) = -\mathcal{S}(t)^{-1}\left(x - \exp(t\mathcal{A})x_0\right)$. Then, Equation (12) can be used to rewrite Equation (8):

$$\mathbb{E}_{\mathbb{Q}}\left[\int_0^T \|\gamma_{\boldsymbol{\theta}}(X_t,t)\|_{R^{\frac{1}{2}}H}^2 dt\right] = \mathbb{E}_{\mathbb{Q}}\left[\int_0^T \|\tilde{\gamma}_{\boldsymbol{\theta}}(X_t, X_0, t)\|_{R^{\frac{1}{2}}H}^2 dt\right] + I, \tag{13}$$

where $I$ is a quantity independent of $\boldsymbol{\theta}$. Knowledge of the conditional true score $D_x \log \rho_t^{(d)}(x \mid x_0)$ and cheap simulation of the forward dynamics, allows for easier numerical optimization than the more general case of $f \neq 0$.

## 3 Sampling theorem for FDPs

The theory of FDPs developed so far is valid for real, separable Hilbert spaces. Our goal now is to specify for which subclass of functions it is possible to perfectly reconstruct the original function given only its evaluation in a countable set of points. We present a generalization of the sampling theorem (Shannon, 1949), which allows us to move from generic Hilbert spaces to a domain which is amenable to a practical implementation of FDPs, and their application to common functional representation of data such as images, data on manifolds, and more. We model these functions as objects belonging to the set of square integrable functions over $C^\infty$ *homogeneous* manifolds $M$ (such as $\mathbb{R}^N, \mathbb{S}^N$, etc...), i.e., the Hilbert space $H = L_2(M)$. Then, exact reconstruction implies that all the relevant information about the considered functions is contained in the set of sampled points.

First, we define functions that are *band-limited*:

**Definition 1.** *A function $x$ in $H = L_2(M)$ is a spectral entire function of exponential type $\nu$ (SE-$\nu$) if $|\Delta^{\frac{k}{2}} x| \leq \nu^k |x|, k \in \mathbb{N}$. Informally, the "Fourier Transform" of $x$ is contained in the interval $[0, \nu]$ (Pesenson, 2000).*

Second, we define grids that cover the manifold with balls, without too much overlap. Those grids will be used to collect the function samples. Their formal definition is as follows:

**Definition 2.** *$Y(r, \lambda)$ denotes the set of all sets of points $Z = \{p_i\}$ such that: i) $\inf_{j \neq i} dist(p_j, p_i) > 0$ and ii) balls $B(p_i, \lambda)$ form a cover of $M$ with multiplicity $< r$.*

Combining the two definitions, we can state the key result of this Section. As long as the sampled function is band-limited, if the samples grid is sufficiently fine, exact reconstruction is possible:

**Theorem 2.** *For any set $Z \in Y(r, \lambda)$, any SE-$\nu$ function $x$ is uniquely determined by its set of values in $Z$ (i.e. $\{x[p_i]\}$) as long as $\lambda < d$, that is*

$$x = \sum_{p_i \in Z} x[p_i] m_{p_i}, \tag{14}$$

*where $m_{p_i} : M \to H$ are known polynomials[2], and the notation $x[p]$ indicates that the function $x$ is evaluated at point $p$.*

A precise definition of the value of the constant $d$ and its interpretation is outside the goal of this work, and we refer the interested reader to Pesenson (2000) for additional details. For our purposes, it is sufficient to interpret the condition in Theorem 2 as a generalization of the classical Shannon-Nyquist sampling theorem (Shannon, 1949). Under this light, Theorem 2 has practical relevance, because it gives the conditions for which the sampled version of functions contains all the information of the original functions. Indeed, given the set of points $p_i$ on which function $x$ is evaluated, it is possible to reconstruct exactly $x[p]$ for arbitrary $p$.

**The uncertainty principle.** It is not always possible to define Hilbert spaces of square integrable functions that are simultaneously homogeneous and separable, for all the manifolds $M$ of interest. In other words, it is difficult in practice to satisfy both the requirements for FDPs to exist, and for the sampling theorem to be valid (see an example in Appendix C). Nevertheless, it is possible to quantify the reconstruction error, and realize that practical applications of FDPs are viable. Indeed, given a compactly supported function $x$, and a set of points $Z$ with *finite* cardinality, we can upper-bound the reconstruction error $\left\| \sum_{p_i \in Z} x[p_i] m_{p_i} - x \right\|_H$ with:

$$\underbrace{\left\| \sum_{p_i \in Z} (x[p_i] - x^\nu[p_i]) m_{p_i} \right\|_H}_{\epsilon_1} + \underbrace{\left\| \sum_{p_i \in Z} x^\nu[p_i] m_{p_i} - x^\nu \right\|_H}_{\epsilon_2} + \underbrace{\|x^\nu - x\|_H}_{\epsilon_3} = \epsilon, \tag{15}$$

where $x^\nu$ is the SE-$\nu$ bandlimited version of $x$, obtained by filtering out – in the frequency domain – any component larger than $\nu$. The error $\epsilon_1$ is due to $x \neq x^\nu$. The term $\epsilon_2$ is the reconstruction error

---

[2]Precisely, they are the limits of spline polynomials that form a Riesz basis for the Hilbert space of polyharmonic functions with singularities in $Z$ (Pesenson, 2000).

due to finiteness of $|Z|$: the sampling theorem applies to $x^\nu$, but the corresponding sampling grid has infinite cardinality. Finally, the term $\epsilon_3$ quantifies the energy omitted by filtering out the frequency components of $x^\nu$ larger than $\nu$. This (loose) upper bound allows us to understand quantitatively the degree to which the sampling theorem does not apply for the cases of interest. Although deriving tighter bounds is possible, this is outside the scope of this work. What suffices is that in many practical cases, when functions are obtained from natural sources, it has been observed that functions are nearly time and bandwidth limited (Slepian, 1983). Consequently, as long as the sampling grid is sufficiently fine, the reconstruction error $\epsilon$ is negligible.

We now hold all the ingredients to formulate generative functional diffusion models using the Hilbert space formalism and *implement* them using a finite grid of points, which is what we do next.

## 4 Score Network Architectural Implementations

We are now equipped with the ELBO (Equation (8)) and a score function $s_{\boldsymbol{\theta}}$ that implements the mapping $H \times [0, T] \times \mathbb{R}^m \to H$. We could then train the score by optimizing the ELBO and produce samples arbitrary close to the true data measure $\rho_0$. However, since the domain of the score function is the infinite-dimensional Hilbert space, such a mapping cannot be implemented in practice. Indeed, having access to samples of functions on finite grid of points is, in general, not sufficient. However, when the conditions for Theorem 2 hold, we can substitute – with no information loss – $x \in H$ with its collection of samples $\{x[p_i], p_i\}$. This allows considering score network architectures that receive as input a collection of points, and not *abstract* functions. Such architectures should be flexible enough to work with an arbitrary number of input samples at arbitrary grid points, and produce as outputs functions in $H$.

### 4.1 Implicit Neural Representation

The first approach we consider in this work is based on the idea of Implicit Neural Representations (INRs) (Sitzmann et al., 2020). These architectures can receive as inputs functions sampled at arbitrary, possibly irregular points, and produce output functions evaluated at any desired point. Unfortunately, the encoding of the inputs is not as straightforward as in the Neural Fourier Operator (NFO) case, and some form of autoencoding is necessary. Note, however, that in traditional score-based diffusion models (Song et al., 2021), the parametric score function can be thought of as a denoising autoencoder. This is a valid interpretation also in our case, as it is evident by observing the term $\mathbb{E}\left[X_0 \mid X_t = x\right]$ of the true score function in Equation (11). Since INRs are powerful denoisers (Kim et al., 2022a), combined with their simple design and small number of parameters, in this Section we discuss how to implement the score network of FDPs using INRs.

We define a *valid* INR as a parametric family $(\boldsymbol{\psi}, t, \boldsymbol{\theta})$ of functions in $H$, i.e., mappings $\mathbb{R}^m \times [0, T] \times \mathbb{R}^m \to H$. A valid INR is the central building block for the implementation of the parametric score function, and it relies on two sets of parameters: $\boldsymbol{\theta}$, which are the parameters of the score function that we optimize according to Equation (8), and $\boldsymbol{\psi}$, which serve the purpose of building a mapping from $H$ into a finite dimensional space. More formally:

**Definition 3.** *Given a manifold $M$, a valid Implicit Neural Representation (*INR*) is an element of $H$ defined by a family of parametric mappings $n(\boldsymbol{\psi}, t, \boldsymbol{\theta})$, with $t \in [0, T], \boldsymbol{\theta}, \boldsymbol{\psi} \in \mathbb{R}^m$. That is, for $p \in M$, we have $n(\boldsymbol{\psi}, t, \boldsymbol{\theta})[p] \in \mathbb{R}$. Moreover, we require $n(\boldsymbol{\psi}, t, \boldsymbol{\theta}) \in L_2(M)$.*

A valid INR as defined in Definition 3 is not sufficiently flexible to implement the parametric score function $s_{\boldsymbol{\theta}}$, as it cannot accept input elements from the infinite-dimensional Hilbert space $H$: indeed, the score function is defined as a mapping over $H \times [0, T] \times \mathbb{R}^m \to H$, whereas the valid INR is a mapping defined over $\mathbb{R}^m \times [0, T] \times \mathbb{R}^m \to H$. Then, we use the second set of parameters $\boldsymbol{\psi}$ to condensate all the information of a generic $x \in H$ into a finite-dimensional vector. When the conditions for Theorem 2 hold, we can substitute — with no information loss — $x \in H$ with its collection of samples $\{x[p_i], p_i\}$. Then, we can construct an implicitly defined mapping $g : H \times [0, T] \times \mathbb{R}^m \to \mathbb{R}^m$ as:

$$g(\{x[p_i], p_i\}, t, \boldsymbol{\theta}) = \arg\min_{\boldsymbol{\psi}} \sum_{p_i} \left( n(\boldsymbol{\psi}, t, \boldsymbol{\theta})[p_i] - x[p_i] \right)^2. \tag{16}$$

In this work, we consider the *modulation* approach to INRs. The set of parameters $\boldsymbol{\psi}$ are obtained by minimizing Equation (16) using few steps of gradient descent on the objective

$\sum_{p_i} (n(\boldsymbol{\psi}, t, \boldsymbol{\theta})[p_i] - x[p_i])^2$, starting from the zero initialization of $\boldsymbol{\psi}$. This approach, also explored by Dupont et al. (2022b), is based on the concept of meta-learning (Finn et al., 2017). In summary, our method constructs mappings $H \times [0, T] \times \mathbb{R}^m \rightarrow H$, where the same INR is used first to encode $x$ into $\boldsymbol{\psi}$, and subsequently to output the value functions for any desired input point $p$, thus implementing the following score network:

$$s_{\boldsymbol{\theta}}(x, t) = -(\mathcal{S}(t))^{-1}\left(x - \exp(t\mathcal{A})n(g(\{x[p_i], p_i\}, t, \boldsymbol{\theta}), t, \boldsymbol{\theta})\right). \tag{17}$$

### 4.2 Transformers

As an alternative approach, we consider implementing the score function $s_{\boldsymbol{\theta}}$ using transformer architectures Vaswani et al. (2017), by interpreting them as mappings between Hilbert spaces (Cao, 2021). We briefly summarize here such a perspective, focusing on a single attention layer for simplicity, and adapt the notation used throughout the paper accordingly.

Consider the space $L_2(M)$, with the usual collection of samples $\{x[p_i], p_i\}$. As a first step, both the "*features*" $\{x[p_i]\}$ and positions $\{p_i\}$ are embedded into some higher dimensional space and summed together, to obtain a sequence of vectors $\{y_i\}$. Then, three different (learnable) matrices $\theta^{(Q)}, \theta^{(K)}, \theta^{(V)}$ are used to construct the linear transformations of the vector sequence $\{y_i\}$ as $\hat{Y}^{(Q)} = \{\hat{y}_i^{(Q)} = \theta^{(Q)}y_i\}, \hat{Y}^{(K)} = \{\hat{y}_i^{(K)} = \theta^{(K)}y_i\}, \hat{Y}^{(V)} = \{\hat{y}_i^{(V)} = \theta^{(V)}y_i\}$. Finally, the three matrices $\hat{Y}^{(Q,K,V)}$ are multiplied together, according to any variant of the attention mechanism. Indeed, different choices for the order of multiplication and normalization schemes in the products and in the matrices correspond to different attention layers Vaswani et al. (2017). In practical implementations, these operations can be repeated multiple times (multiple attention layers) and can be done in parallel according to multiple projection matrices (multiple heads).

The perspective explored in (Cao, 2021) is that it is possible to interpret the sequences $\hat{y}_i^{(Q,K,V)}$ as **learnable** basis functions in some underlying *latent* Hilbert space, evaluated at the set of coordinates $\{p_i\}$. Furthermore, depending on the type of attention mechanism selected, the operation can be interpreted as a different mapping between Hilbert spaces, such as Fredholm equations of the second-kind or Petrov–Galerkin-type projections (Cao (2021) Eqs. 9 and 14).

While a complete treatment of such an interpretation is outside the scope of this work, what suffices is that it is possible to claim that transformer architectures are a viable candidate for the implementation of the desired mapping $H \times [0, T] \times \mathbb{R}^m \rightarrow H$, a possibility that we explore experimentally in this work. It is worth noticing that, compared to the approach based on INRs, resolution invariance is only *learned*, and not guaranteed, and that the number of parameters is generally higher compared to an INR. Nevertheless, learning the parameters of transformer architectures does not require meta-learning, which is a practical pain-point of INRs used in our context. Additional details for the transformer-based implementation of the score network are available in Appendix E.

Finally, for completeness, it is worth mentioning that a related class of architectures, the Neural Operators and NFOs (Kovachki et al., 2021; Li et al., 2020), are also valid alternatives. However, such architectures require the input grid to be regularly spaced (Li et al., 2020), and their output function is available only at the same points $p_i$ of the input, which would reduce the flexibility of FDPs.

## 5 Training and sampling of FDPs

Given the parametric score function $s_{\boldsymbol{\theta}}$ from Equation (17), by simulating the reverse FDP, we generate samples whose statistical measure $\chi_0$ is close in KL sense to $\rho_0$. Next, we explain how to numerically compute of the quantities in Equation (13), which is part of the ELBO in Equation (8), and how to generate new samples from the trained FDP (simulation of Equation (10)).

**ELBO Computation.** Equation (8) involves Equation (13), which requires the computation of the Hilbert space norm. The grid of points $x[p_i]$ is interpolated in $H$ as $\sum_i x[p_i]\xi^i$. Then, the norm of interest can be computed as:

$$\left\|\sum_i x[p_i]\xi^i\right\|_{R^{\frac{1}{2}}H}^2 = \langle R^{-\frac{1}{2}}\sum_i x[p_i]\xi^i, R^{-\frac{1}{2}}\sum_i x[p_i]\xi^i\rangle_H = \sum_{k=1}^{\infty}(r^k)^{-1}\left(\left\langle\sum_{i=1}^{N} x[p_i]\xi^i, e^k\right\rangle\right)^2. \tag{18}$$

Depending on the choice of $\xi^i, e^i$, the sum w.r.t the index $k$ is either naturally truncated or it needs to be further approximated by selecting a cutoff index value. Finally, training can then be performed by minimizing:

$$\mathbb{E}_{\mathbb{Q}}\left[\int_0^T \|\tilde{\gamma}_{\boldsymbol{\theta}}(X_t, t)\|^2_{R^{\frac{1}{2}}H}\,\mathrm{d}t\right] \simeq \mathbb{E}_{\sim(20)}\left[\int_0^T \sum_{k=1}^\infty (r^k)^{-1}\left(\left\langle \sum_{i=1}^N \left(\tilde{\gamma}_{\boldsymbol{\theta}}(\sum_i X_t[p_i]\xi^i, t)[p_i]\right)\xi^i, e^k\right\rangle\right)^2 \mathrm{d}t\right]. \tag{19}$$

**Numerical integration.** Simulation of infinite dimensional SDEs is a well studied domain (Debussche, 2011), including finite difference schemes (Gyöngy, 1998, 1999; Yoo, 2000), finite element methods and/or Galerkin schemes (Hausenblas, 2003a,b; Shardlow, 1999). In this work, we adopt a finite element approximate scheme, and introduce the *interpolation* operator, from $\mathbb{R}^{|Z|}$ to $H$, i.e. $\sum_i x[p_i]\xi^i$ (Hausenblas, 2003b). Notice that, in general, the functions $\xi^i$ differ from the basis $e^i$. In addition, the *projection* operator maps functions from $H$ into $\mathbb{R}^L$, as $\langle x, \zeta^j\rangle, \zeta^j \in H$. Usually, $L = |Z|$. When $\zeta^i = \xi^i$ the scheme is referred to as the Galerkin scheme. We consider instead a point matching scheme (Hausenblas, 2003b), in which $\zeta^i = \delta[p - p_i]$ with $\delta$ in Dirac sense, and consequently $\langle x, \zeta^i\rangle = x[p_i]$. Then, the infinite dimensional SDE of the forward process from Equation (9) is approximated by the finite ($|Z|$) dimensional SDE:

$$\mathrm{d}X_t[p_k] = \left(\left\langle \mathcal{A}\sum_i X_t[p_i]\xi^i, \zeta^k\right\rangle\right)\mathrm{d}t + \mathrm{d}W_t[p_k], \quad k = 1, \ldots, |Z|. \tag{20}$$

Similarly, the reverse process described by Equation (10) corresponds to the following SDE:

$$\mathrm{d}\hat{X}_t[p_k] = \left(-\left\langle \mathcal{A}\sum_i \hat{X}_t[p_i]\xi^i, \zeta^k\right\rangle + \left\langle Rs_{\boldsymbol{\theta}}(\sum_i \hat{X}_t[p_i]\xi^i, T - t), \zeta^k\right\rangle\right)\mathrm{d}t + \mathrm{d}\hat{W}_t[p_k], \tag{21}$$
$$k = 1, \ldots, |Z|.$$

Equation (21) is a finite dimensional SDE, and consequently we can use any known numerical integrator to simulate its paths. In Appendix D we provide a complete toy example to illustrate our approach in a simple scenario, where we emphasize practical choices.

## 6    Experiments

Despite a rather involved theoretical treatment, the implementation of FDPs is simple. We implemented our approach in JAX (Bradbury et al., 2018), and use WANDB (Biewald, 2020) for our experimental protocol. Additional details on implementation, and experimental setup, as well as more experiments are available in Appendix E.

We evaluate our approach on image data, using the CELEBA $64 \times 64$ (Liu et al., 2015) dataset. Our comparative analysis with the state-of-the-art includes generative quality, using the FID score (Heusel et al., 2017), and parameter count for the score network. We also discuss (informally) the complexity of the network architecture, as a measure of the engineering effort in exploring the design space of the score network. We compare against vanilla Score Based Diffusion (SBD) (Song et al., 2021), From Data To Functa (FD2F) (Dupont et al., 2022a) which diffuses latent variables obtained from an INR, Infinite Diffusion ($\infty$-DIFF) (Bond-Taylor & Willcocks, 2023), which is a recent approach that is only partially suited for the functional domain, as it relies on the combination of Fourier Neural Operators and a classical convolutional U-NET backbone. Our FDP method is implemented using either MLP or Transformers. In the first case, we consider a score network implemented as a simple MLP with 15 layers and 256 neurons in each layer. The activation function is a Gabor wavelet activation function (Saragadam et al., 2023). In the latter case, our approach is built upon the UViT backbone as detailed by Bao et al. (2022). The architecture comprises 7 layers, with each layer composed of a self-attention mechanism with 8 attention heads and a feedforward layer.

We present quantitative results in Table 1, showing that our method **FDP(MLP)** achieves an impressively low FID score, given the extremely low parameter count, and the simplicity of the architecture. FD2F obtains a worse (larger) FID score, while having many more parameters, due to the complex

parametrization of their score network. As a reference we report the results of SBD, where the price to be pay to achieve an extremely low FID is to have many more parameters and a much more intricate architecture. Finally, the very recent ∞-DIFF method, has low FID-CLIP score (Kynkäänniemi et al., 2022), but requires a very complex architecture and more than 2 orders of magnitude more parameters than our approach. Showcasing the flexibility of the proposed methodology, we consider a more complex architecture based on Vision Transformers (**FDP(UViT)**). These corresponding results indicate improvements in terms of image quality (FID score=11) and do not require meta-learning steps, but require more parameters (O(20M)) than the INR variant. To the best of our knowledge, none of related work in the purely functional domain (Lim et al., 2023; Hagemann et al., 2023; Dutordoir et al., 2022; Kerrigan et al., 2022) provides results going beyond simple data-sets. Finally, we present some qualitative results in Figures 1 and 2 clearly showing that the proposed methodology is capable of producing diverse and detailed images.

| Methods | FID ($\downarrow$) | FID-CLIP ($\downarrow$) | Params |
|---|---|---|---|
| **FDP(MLP)** | 35.00 | 12.44 | $O$(1 M) |
| **FDP(UViT)** | 11.00 | 6.55 | $O$(20 M) |
| FD2F | 40.40 | - | $O$(10 M) |
| SBD | 3.30 | - | $O$(100 M) |
| ∞-DIFF | - | 4.57 | $O$(100 M) |

**Table 1:** Quantitative results, CELEBA data-set. (FID-CLIP (Kynkäänniemi et al., 2022))

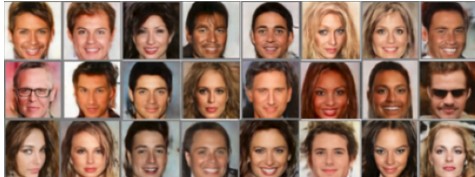

**Figure 1:** Qualitative results with MLP.

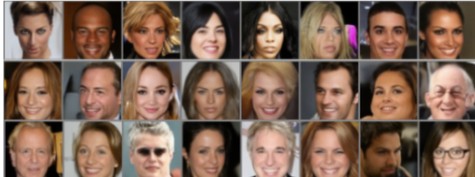

**Figure 2:** Qualitative results with UViT.

# 7 Conclusion, Limitations and Broader Impact

We presented a theoretical framework to define functional diffusion processes for generative modeling. FDPs generalize traditional score-based diffusion models to infinite-dimensional function spaces, and in this context we were the first to provide a full characterization of forward and backward dynamics, together with a formal derivation of an ELBO that allowed the estimation of the parametric score function driving the reverse dynamics.

To use FDPs in practice, we carefully studied for which subset of functions it was possible to operate on a countable set of samples without losing information. We then proceeded to introduce a series of methods to jointly model – using only a simple INR or a Transformer – an approximate functional representation of data on discrete grids, and an approximate score function. Additionally, we detailed practical training procedures of FDPs, and integration schemes to generate new samples.

The implementation of FDPs for generative modeling was simple. We validated the viability of FDPs through a series of experiments on real images, where we show, while only using a simple MLP for learning the score network, extremely promising results in terms of generation quality.

Like other works in the literature, the proposed method can have both positive (e.g., synthesizing new data automatically) and negative (e.g., deep fakes) impacts on society depending on the application.

# 8 Acknowledgments

GF gratefully acknowledges support from Huawei Paris and the European Commission (ADROIT6G Grant agreement ID: 101095363). MF gratefully acknowledges support from the AXA Research Fund and the Agence Nationale de la Recherche (grant ANR-18-CE46-0002 and ANR-19-P3IA-0002).

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

# Supplementary Material: Continuous-Time Functional Diffusion Processes

# A   Reverse Functional Diffusion Processes

In this Section, we review the mathematical details to obtain the backward FDP discussed in Theorem 1. Depending on the considered class of noise, different approaches are needed. First, we present in Appendix A.1 the conditions to ensure existence of the backward process , which we use if the $C$ operator is an identity matrix, $C = I$. Then we move to a different approach in Appendix A.2 for the case $C \neq I$.

## A.1   Follmer Formulation

The work in Föllmer (1986) is based on a finite entropy condition, which we report here as Condition 1. One simple way to ensure that the condition is satisfied is to assume:

**Condition 1.** *For a given $k$, define $\mathbb{Q}_{(k)}$ to be the path measure corresponding to the (infinite) system*

$$\begin{cases} \mathrm{d}X_t^i = b^i(X_t, t)\mathrm{d}t + \mathrm{d}W_t^i, & i \neq k \\ \mathrm{d}X_t^i = \mathrm{d}W_t^k, & i = k. \end{cases} \tag{22}$$

*We say that $\mathbb{Q}$ satisfies the finite local entropy condition if* $\mathrm{KL}\left[\mathbb{Q} \parallel \mathbb{Q}_{(k)}\right] < \infty, \forall k$.

Define $\mathcal{F}_t^{(i)} = \sigma(X_0^i, X_s^j, 0 \leq s \leq t, j \neq i)$.

**Assumption 1.**

$$\int_0^T b^i(X_t, t)^2 \mathrm{d}t + \sum_{j \neq i} \mathbb{E}\left[\int_0^T \left(b^j(X_t, t) - \mathbb{E}\left[b^j(X_t, t) \,|\, \mathcal{F}_t^{(i)}\right]\right)^2 \mathrm{d}t\right] < \infty, \mathbb{Q}_{(i)} a.s. \tag{23}$$

Notice that if Assumption 1 is true, then Condition 1 holds (Föllmer (1986), Thm. 2.23)

**Theorem 3.** *If* $\mathrm{KL}\left[\mathbb{Q} \parallel \mathbb{Q}_{(k)}\right] < \infty$, *then* $\mathrm{KL}\left[\hat{\mathbb{Q}} \parallel \hat{\mathbb{Q}}_{(k)}\right] < \infty$.

*Proof.* The proof can be obtained by adapting the result of Lemma 3.6 of Föllmer & Wakolbinger (1986). □

This Theorem states that if the forward FDP path measure $\mathbb{Q}$ satisfies the finite local entropy condition, then also the reverse FDP path measure $\hat{\mathbb{Q}}$ satisfies the finite local entropy condition.

**Theorem 4.** *Let $\mathbb{Q}$ be a finite entropy measure. Then:*

$$\begin{cases} \mathrm{d}X_t^k = b^k(X_t, t)\mathrm{d}t + \mathrm{d}W_t^k, & under \quad \mathbb{Q} \\ \mathrm{d}\hat{X}_t^k = \hat{b}^k(\hat{X}_t, t)\mathrm{d}t + \mathrm{d}\hat{W}_t^k, & under \quad \hat{\mathbb{Q}} \end{cases} \tag{24}$$

*where:*

$$\frac{\partial \log\left(\rho_t^{(d)}(x^k \,|\, x^j, j \neq k)\right)}{\partial x^k} = \hat{b}^k(x, T - t) + b^k(x, t) \tag{25}$$

*Proof.* For the proof, we refer to Theorem 3.14 of Föllmer & Wakolbinger (1986). □

## A.2   Millet Formulation

Let $L^2(R) = \{x \in H : \sum r^i(x^i)^2 < \infty\}$. For simplicity, we overload the notation of the letter $K$, and use it for generic constants, that might be different on a case by case basis.

**Assumption 2.**

$$\forall x \in L^2(R), \sup_t\{\sum r^i(b^i(x, t))^2\} + \sum(r^i)^2 \leq K(1 + \sum r^i(x^i)^2)$$

$$\forall x, y \in L^2(R), \sup_t\{\sum r^i(b^i(x, t) - b^i(y, t))^2\} \leq K \sum r^i(x^i - y^i)^2$$

This assumption is simply the translation of H1 from Millet et al. (1989) to our notation.

**Assumption 3.** *There exists an increasing sequence of finite subsets $J(n), n \in \mathrm{N}, \cup_n J(n) = \mathrm{N}$ such that $\forall n \in \mathrm{N}, M > 0$ there exists a constant $K(M, n)$ such that the following holds:*

$$\sup_t \left( \sup_{i \in J(n)} \left( \left( \sup_x |b^i(x, t)| : \sup_{j \in J(n)} |x^j| \leq M \right) + \sum_j r^j \right) \right) \leq K(M, n).$$

Again, this assumption is simply the translation of H5 from Millet et al. (1989) to our notation.

**Assumption 4.** *Either i):*

$$\forall x, y \in L^2(R), \sup_t \{ \sum r^i (b^i(x, t) - b^i(y, t))^2 \} \leq K \sum (r^i)^2 (x^i - y^i)^2,$$

*or ii): $\forall i, b^i(x)$ is a function of $x$ for at most $M$ coordinates and*

$$\forall x, y \in L^2(R), \sup_t \{ \sum (r^i)^2 (b^i(x, t) - b^i(y, t))^2 \} \leq K \sum (r^i)^2 (x^i - y^i)^2.$$

This corresponds to satisfying either H3 or jointly H2 and H4 of Millet et al. (1989). For simplicity, we can combine together the different assumptions into

**Assumption 5.** *Let Assumption 2, Assumption 3, and Assumption 4 hold.*

Finally, we state required assumptions about the density:

**Assumption 6.** *Suppose that the initial condition is $X_0 \in L^2(R)$.*

- *Assume that the conditional law of $x^i$ given $x^j, j \neq i$ has density $\rho_t^{(d)}(x^i \,|\, x^j, j \neq i)$ w.r.t Lebesgue measure on $\mathbb{R}$.*

- *Assume that $\int_{t_0}^1 \int_{D_J} |r^i \frac{\mathrm{d}}{\mathrm{d}x^i}(\rho_t^{(d)}(x^i \,|\, x^j, j \neq i))| \mathrm{d}x^i \rho_t(\mathrm{d}x^{j \neq i}) \mathrm{d}t < \infty$, for fixed subset $J \subset \mathrm{N}, t_0 > 0$ and $D_J = \{ (\prod_{j \in J} K_j) \times (\prod_{j \notin J} \mathbb{R}), K_j \text{ compact in } \mathbb{R} \} \cap L^2(R)$.*

We reported in our notation the content of Theorem 4.3 of Millet et al. (1989). This can be used to prove the existence of the backward process.

## A.3 Proof of Theorem 1

If $R = I$, then we assume Assumption 1. Consequently, $\mathbb{Q}$ is a finite entropy measure. Then Theorem 4 holds, from which the desired result. If, instead $R \neq I$, then we require Assumption 5, Assumption 6. Application of Thm 4.3 of Millet et al. (1989) allows to prove the validity of Theorem 1 also in this case.

### A.3.1 Proof of Corollary 1

Assumption 5 is required directly. We need to show that with the considered restrictions Assumption 6 is valid.

Since $\sum_i r^i < \infty$, then $\sum_i (r^i)^2 = K_a < \infty$. Moreover, $(b^i(x^i, t))^2 < K_b^2(x^i)^2$. Then, $\forall x \in L^2(R)$, the following holds $\sup_t \{ \sum r^i (b^i(x, t))^2 \} + \sum (r^i)^2 \leq \sum r^i K_b^2(x^i)^2 + K_a \leq \max(K_a, K_b^2) \left( 1 + \sum r^i(x^i)^2 \right)$. Similarly, $\forall x, y \in L^2(R)$ we have $\sup_t \{ \sum r^i (b^i(x, t) - b^i(y, t))^2 \} \leq \sum r^i K_b^2 (x^i - y^i)^2$. Thus Assumption 2 is satisfied.

Since $b^i(x, t)$ is bounded and independent on $t$, Assumption 3 is satisfied, as explicitly discussed in Millet et al. (1989).

Finally, since $b^i(x)$ is a function of $x$ for $M = 1$ coordinate, and $\sup_t \{ \sum (r^i)^2 (b^i(x, t) - b^i(y, t))^2 \} \leq \sum (r^i)^2 K_b^2 (x^i - y^i)^2$, Assumption 4 is satisfied.

Then, combined toghether Assumption 5 holds.

## A.4 Girsanov Regularity

**Condition 2.** *Assume that $\gamma_{\boldsymbol{\theta}}(x,t)$ is an $\hat{\mathcal{F}}$ measurable process and that either:*

$$\mathbb{E}_{\hat{\mathbb{Q}}}\left[\exp\left(\frac{1}{2}\int_0^T \left\|\gamma_{\boldsymbol{\theta}}(\hat{X}_t,t)\right\|^2_{R^{\frac{1}{2}}H}\mathrm{d}t\right)\right] = \mathbb{E}_{\mathbb{Q}}\left[\exp\left(\frac{1}{2}\int_0^T \left\|\gamma_{\boldsymbol{\theta}}(X_t,t)\right\|^2_{R^{\frac{1}{2}}H}\mathrm{d}t\right)\right] < \infty, \quad (26)$$

*or*

$$\exists \delta > 0 : \mathbb{E}_{\hat{\mathbb{Q}}}\left[\exp\left(\frac{1}{2}\left\|\gamma_{\boldsymbol{\theta}}(\hat{X}_\delta,\delta)\right\|_{R^{\frac{1}{2}}H}\mathrm{d}t\right)\right] < \infty. \quad (27)$$

This is equivalent to the regularity condition in eq. 10.23 of Da Prato & Zabczyk (2014) or Proposition 10.17 in Da Prato & Zabczyk (2014).

## A.5 Proof of KL divergence expression

We leverage Equation (7) to express the Kullback-Leibler divergence as:

$$\mathrm{KL}\left[\hat{\mathbb{Q}} \,\|\, \hat{\mathbb{P}}^{\chi_T}\right] = \mathbb{E}_{\hat{\mathbb{Q}}}\left[\log\frac{\mathrm{d}\hat{\mathbb{Q}}_0}{\mathrm{d}\hat{\mathbb{P}}_0} + \log\frac{\mathrm{d}\rho_T}{\mathrm{d}\chi_T}\right] = \mathbb{E}_{\hat{\mathbb{Q}}}\left[\log\frac{\mathrm{d}\hat{\mathbb{Q}}_0}{\mathrm{d}\hat{\mathbb{P}}_0}\right] + \mathrm{KL}\left[\rho_T \,\|\, \chi_T\right] =$$

$$\mathbb{E}_{\hat{\mathbb{Q}}}\left[-\int_0^T \langle\gamma_{\boldsymbol{\theta}}(\hat{X}_t,t),\mathrm{d}\hat{W}_t\rangle_{R^{\frac{1}{2}}H} + \frac{1}{2}\int_0^T \left\|\gamma_{\boldsymbol{\theta}}(\hat{X}_t,t)\right\|^2_{R^{\frac{1}{2}}H}\mathrm{d}t\right] + \mathrm{KL}\left[\rho_T \,\|\, \chi_T\right] =$$

$$\frac{1}{2}\mathbb{E}_{\hat{\mathbb{Q}}}\left[\int_0^T \left\|\gamma_{\boldsymbol{\theta}}(\hat{X}_t,t)\right\|^2_{R^{\frac{1}{2}}H}\mathrm{d}t\right] + \mathrm{KL}\left[\rho_T \,\|\, \chi_T\right] = \frac{1}{2}\mathbb{E}_{\mathbb{Q}}\left[\int_0^T \|\gamma_{\boldsymbol{\theta}}(X_t,t)\|^2_{R^{\frac{1}{2}}H}\mathrm{d}t\right] + \mathrm{KL}\left[\rho_T \,\|\, \chi_T\right].$$

Moreover, since

$$\mathrm{KL}\left[\hat{\mathbb{Q}} \,\|\, \hat{\mathbb{P}}^{\chi_T}\right] = \mathbb{E}_{\mathbb{Q}}\left[\log\frac{\mathrm{d}\hat{\mathbb{Q}}_T}{\mathrm{d}\hat{\mathbb{P}}_T^{\chi_T}} + \log\frac{\mathrm{d}\rho_0}{\mathrm{d}\chi_0}\right] \geq \mathrm{KL}\left[\rho_0 \,\|\, \chi_0\right],$$

we can combine the two results and obtain Equation (8)

## A.6 Conditional score matching

In this subsection we prove the equality in Equation (13):

$$\mathbb{E}_{\mathbb{Q}}\left[\int_0^T \|\gamma_{\boldsymbol{\theta}}(X_t,t)\|^2_{R^{\frac{1}{2}}H}\mathrm{d}t\right] = \int_0^T\int_H \|\gamma_{\boldsymbol{\theta}}(x,t)\|^2_{R^{\frac{1}{2}}H}\mathrm{d}t\mathrm{d}\rho_t(x) =$$

$$\int_0^T\int_H \left\|D_x\log\rho_{T-t}^{(d)}(x) - s_{\boldsymbol{\theta}}(x,T-t)\right\|^2_{R^{\frac{1}{2}}H}\mathrm{d}t\mathrm{d}\rho_t(x) =$$

$$\int_0^T\int_{H\times H} \left\|D_x\log\rho_t^{(d)}(x) - s_{\boldsymbol{\theta}}(x,t)\right\|^2_{R^{\frac{1}{2}}H}\mathrm{d}t\mathrm{d}\rho_t(x,x_0) =$$

$$\int_0^T\int_{H\times H} \left\|D_x\log\rho_t^{(d)}(x) - D_x\log\rho_t^{(d)}(x\,|\,x_0) + D_x\log\rho_t^{(d)}(x\,|\,x_0) - s_{\boldsymbol{\theta}}(x,t)\right\|^2_{R^{\frac{1}{2}}H}\mathrm{d}t\mathrm{d}\rho_t(x,x_0) =$$

$$\int_0^T\int_{H\times H} \left\|D_x\log\rho_t^{(d)}(x) - D_x\log\rho_t^{(d)}(x\,|\,x_0)\right\|^2_{R^{\frac{1}{2}}H} + \left\|D_x\log\rho_t^{(d)}(x\,|\,x_0) - s_{\boldsymbol{\theta}}(x,t)\right\|^2_{R^{\frac{1}{2}}H} +$$

$$2\left\langle D_x\log\rho_t^{(d)}(x) - D_x\log\rho_t^{(d)}(x\,|\,x_0), D_x\log\rho_t^{(d)}(x\,|\,x_0) - s_{\boldsymbol{\theta}}(x,t)\right\rangle\mathrm{d}t\mathrm{d}\rho_t(x,x_0).$$

To simplify the equality, we need to notice that:

$$\rho_t^{(d)}(x^i|x^{j\neq i})\mathrm{d}x^i = \mathrm{d}\rho_t(x^i|x^{j\neq i}) = \int_{x_0} \mathrm{d}\rho_t(x_0|x)\mathrm{d}\rho_t(x^i|x^{j\neq i}) = \int_{x_0} \mathrm{d}\rho_t(x^i, x_0|x^{j\neq i}) =$$

$$\int_{x_0} \mathrm{d}\rho_t(x^i|x_0, x^{j\neq i})\mathrm{d}\rho_t(x_0|x^{j\neq i}) = \mathrm{d}x^i \int_{x_0} \rho_t^{(d)}(x^i|x_0, x^{j\neq i})\mathrm{d}\rho_t(x_0|x^{j\neq i}).$$

Then, computing

$$\int_{x_0} \frac{\mathrm{d}}{\mathrm{d}x^i} \log \rho^{(d)}(x^i|x^{j\neq i}, x_0)\mathrm{d}\rho_t(x, x_0) = \int_{x_0} \frac{\frac{\mathrm{d}}{\mathrm{d}x^i}\rho^{(d)}(x^i|x^{j\neq i}, x_0)}{\rho^{(d)}(x^i|x^{j\neq i}, x_0)}\mathrm{d}\rho_t(x, x_0) =$$

$$\int_{x_0} \frac{\frac{\mathrm{d}}{\mathrm{d}x^i}\rho^{(d)}(x^i|x^{j\neq i}, x_0)}{\rho^{(d)}(x^i|x^{j\neq i}, x_0)}\mathrm{d}\rho_t(x^i|x^{j\neq i}, x_0)\mathrm{d}\rho_t(x_0, x^{j\neq i}) = \int_{x_0} \frac{\mathrm{d}}{\mathrm{d}x^i}\rho^{(d)}(x^i|x^{j\neq i}, x_0)\mathrm{d}x^i\mathrm{d}\rho_t(x_0, x^{j\neq i}) =$$

$$\int_{x_0} \frac{\mathrm{d}}{\mathrm{d}x^i}\rho^{(d)}(x^i|x^{j\neq i}, x_0)\mathrm{d}x^i\mathrm{d}\rho_t(x_0|x^{j\neq i})\mathrm{d}\rho_t(x^{j\neq i}) = \frac{\mathrm{d}}{\mathrm{d}x^i}\left(\int_{x_0}\rho^{(d)}(x^i|x^{j\neq i}, x_0)\mathrm{d}\rho_t(x_0|x^{j\neq i})\right)\mathrm{d}x^i\mathrm{d}\rho_t(x^{j\neq i}) =$$

$$\frac{\mathrm{d}}{\mathrm{d}x^i}\rho_t^{(d)}(x^i|x^{j\neq i})\mathrm{d}x^i\mathrm{d}\rho_t(x^{j\neq i}) = \frac{\mathrm{d}\log\rho_t^{(d)}(x^i|x^{j\neq i})}{\mathrm{d}x^i}\rho_t^{(d)}(x^i|x^{j\neq i})\mathrm{d}x^i\mathrm{d}\rho_t(x^{j\neq i}) = \frac{\mathrm{d}\log\rho_t^{(d)}(x^i|x^{j\neq i})}{\mathrm{d}x^i}\mathrm{d}\rho_t(x)$$

Consequently:

$$\int_{H\times H} \left\langle D_x \log \rho_t^{(d)}(x) - D_x \log \rho_t^{(d)}(x\,|\,x_0), s_{\boldsymbol{\theta}}(x, t)\right\rangle \mathrm{d}\rho_t(x, x_0) = 0.$$

Combining together and rearranging the terms, we get the desired Equation (13).

### A.7 Explicit expression of score function

As mentioned in the text, we consider the case $f = 0$. In this case, there exists a weak solution to Equation (1) as:

$$X_t = \exp(t\mathcal{A})X_0 + \int_0^t \exp((t - s)\mathcal{A})\mathrm{d}W_s. \tag{28}$$

Consequently, the true score function has expression:

$$\frac{\mathrm{d}}{\mathrm{d}x^i}\log\rho_t^{(d)}(x^i|x^{j\neq i}) = \frac{\frac{\mathrm{d}}{\mathrm{d}x^i}\rho_t^{(d)}(x^i|x^{j\neq i})}{\rho_t^{(d)}(x^i|x^{j\neq i})} = \frac{\frac{\mathrm{d}}{\mathrm{d}x^i}\int_{x_0}\rho_t^{(d)}(x^i|x_0,x^{j\neq i})\mathrm{d}\rho_t(x_0|x^{j\neq i})}{\rho_t^{(d)}(x^i|x^{j\neq i})} =$$

$$\frac{-\int_{x_0}(s^i)^{-1}\left(x^i-\exp(tb^i)x_0^i\right)\rho_t^{(d)}(x^i|x_0,x^{j\neq i})\mathrm{d}\rho_t(x_0|x^{j\neq i})}{\rho_t^{(d)}(x^i|x^{j\neq i})} =$$

$$\frac{-(s^i)^{-1}\left(x^i\rho_t^{(d)}(x^i|x^{j\neq i})-\int_{x_0}\exp(tb^i)x_0^i\rho_t^{(d)}(x^i|x_0,x^{j\neq i})\mathrm{d}\rho_t(x_0|x^{j\neq i})\right)}{\rho_t^{(d)}(x^i|x^{j\neq i})} =$$

$$\frac{-(s^i)^{-1}\left(x^i\rho_t^{(d)}(x^i|x^{j\neq i})-\int_{x_0}\exp(tb^i)x_0^i\rho_t^{(d)}(x^i|x_0,x^{j\neq i})\mathrm{d}\rho_t(x_0|x^{j\neq i})\right)}{\rho_t^{(d)}(x^i|x^{j\neq i})} =$$

$$\frac{-(s^i)^{-1}\left(x^i\rho_t^{(d)}(x^i|x^{j\neq i})-\int_{x_0^i}\exp(tb^i)x_0^i\rho_t^{(d)}(x^i|x_0^i,x^{j\neq i})\mathrm{d}\rho_t(x_0^i|x^{j\neq i})\right)}{\rho_t^{(d)}(x^i|x^{j\neq i})} =$$

$$\frac{-(s^i)^{-1}\left(x^i\rho_t^{(d)}(x^i|x^{j\neq i})-\int_{x_0^i}\exp(tb^i)x_0^i\rho_t^{(d)}(x^i|x_0^i,x^{j\neq i})\rho^{(d)}(x_0^i|x^{j\neq i})\mathrm{d}x_0^i\right)}{\rho_t^{(d)}(x^i|x^{j\neq i})} =$$

$$\frac{-(s^i)^{-1}\left(x^i\rho_t^{(d)}(x^i|x^{j\neq i})-\int_{x_0^i}\exp(tb^i)x_0^i\rho_t^{(d)}(x^i,x_0^i|x^{j\neq i})\mathrm{d}x_0^i\right)}{\rho_t^{(d)}(x^i|x^{j\neq i})} =$$

$$\frac{-(s^i)^{-1}\left(x^i\rho_t^{(d)}(x^i|x^{j\neq i})-\int_{x_0^i}\exp(tb^i)x_0^i\rho_t^{(d)}(x_0^i|x)\mathrm{d}x_0^i\right)\rho_t^{(d)}(x^i|x^{j\neq i}))}{\rho_t^{(d)}(x^i|x^{j\neq i})} =$$

$$-(s^i)^{-1}\left(x^i-\int_{x_0^i}\exp(tb^i)x_0^i\rho_t^{(d)}(x_0^i|x)\mathrm{d}x_0^i\right)$$

where $s^i = r^i\frac{\exp(2b^it)-1}{2b^i}$. This is exactly the desired Equation (11). Similar calculations allow to prove $D_x\log\rho_t^{(d)}(x\,|\,x_0) = -\mathcal{S}(t)^{-1}\left(x-\exp(t\mathcal{A})x_0\right)$.

## B  Fokker Planck equation

In this Section we discuss the infinite dimensional generalization of the classical Fokker Planck equation. We can associate to Eq. (1) the differential operator:

$$\mathcal{L}_0u(x,t) = D_tu(x,t) + \underbrace{\frac{1}{2}\operatorname{Tr}\{RD_x^2u(x,t)\} + \langle\mathcal{A}x+f(x,t),D_xu(x,t)\rangle}_{\mathcal{L}u(x,t)}, \quad x\in H, t\in[0,T],$$

(29)

where $D_t$ is the time derivative, $D_x, D_x^2$ are first and second order Fréchet derivatives in space. The domain of the operator $\mathcal{L}_0$ is $D(\mathcal{L}_0)$, the linear span of real parts of functions $u_{\phi,h} = \phi(t)\exp(i\langle x,h(t)\rangle), x\in H, t\in[0,T]$ where $\phi\in C^1([0,T]), \phi(T)=0, h\in C^1([0,T];D(\mathcal{A}^\dagger))$, where † indicates adjoint. Provided appropriate conditions are satisfied, see for example Bogachev et al. (2009, 2011), the time varying measure $\rho_t(\mathrm{d}x)\mathrm{d}t$ exists, is unique, and solves the Fokker-Planck equation $\mathcal{L}_0^\dagger\rho_t = 0$.

## C  Uncertainty principle

We here clarify that Hilbert spaces of square integrable functions that are not, in general, simultaneously homogeneous and separable. For example, while $\mathbb{R}$ is homogeneous, the set of square integrable functions over $\mathbb{R}$ is not separable, since the basis is the *uncountable* set $\cos(2\pi\nu p), \sin(2\pi\nu p), \nu\in\mathbb{R}$. Then, FDP requirements are not met, as we need a countable basis. Moreover, we would need in

general an infinite number of samples (grid over the whole $\mathbb{R}$) to reconstruct the functions. Conversely, a set like the interval $I = [0,1] \subset \mathbb{R}$ has *countable* basis $\cos(2\pi tp), \sin(2\pi tp), t \in \mathbb{Z}$ (and thus is separable) and, considering $x$ to be band-limited, a sampling grid with finite cardinality would allow to reconstruct of the function. However, $I$ is not homogeneous as no isometry group exists. Consequently, Theorem 2 is not applicable. To fix the issue, one could naively think of extending any function defined over $I$ to the whole $\mathbb{R}$ by considering $\bar{x}[p] = x[p], p \in I$ and $\bar{x}[p] = 0, p \notin I$. Obviously, if $x \in L_2(I)$ then $\bar{x} \in L_2(\mathbb{R})$. However, since $\bar{x}$ has finite support, it cannot be bandlimited, making such an approach not a viable solution. In classical signal processing literature, the problem is usually referred to as the *uncertainty principle* (Slepian, 1983).

## D A complete example

We present an example in which we cast Equation (20) for square integrable functions over the interval $I = [0,1]$, $L^2(I)$. In this case, one natural selection for the basis is the Fourier basis[3] $e^k = \{\ldots, \exp(-j2\pi 2p), \exp(-j2\pi p), 1, \exp(j2\pi p), \exp(j2\pi 2p), \ldots\}$. Assume the operator $\mathcal{A}$ to be a pseudo-differantial operator, such that $\langle \mathcal{A}x, e^k \rangle = b^k x^k$. Also, assume that $b^k, r^k$ are selected such that conditions of Corollary 1 are met, and consequently the backward process exists. Since we are working with samples collected on the grid $x\,[i/N]$ we first map the samples to the frequency domain, and then build a Fourier-like representation with a finite set of sinusoids. We then define the mapping $\mathfrak{F}(z^i)^k \overset{\text{def}}{=} \sum_{i=0}^{N-1} z^i \exp\!\left(-j2\pi k \frac{i}{N}\right)$ and its inverse $\mathfrak{I}(z^i)^k \overset{\text{def}}{=} N^{-1} \sum_{i=0}^{N-1} z^i \exp\!\left(j2\pi k \frac{i}{N}\right)$. This suggests to consider the following expression for the interpolating functions:

$$\xi^i = \frac{1}{N} \sum_{k=0}^{N-1} e^k \exp\!\left(-j2\pi k \frac{i}{N}\right) = \frac{1}{N} \sum_{k=0}^{N-1} \exp\!\left(j2\pi k\!\left(p - \frac{i}{N}\right)\right).$$

Those functions are indeed nothing but a frequency truncated version of the sinc function, which is the classical reconstruction function of the sampling theorem on 1-D signals. Moreover $\langle \xi^i, \zeta^k \rangle = \delta(i - k)$. We are now ready to show *i)* the expression of the forward process, *ii)* the expression of the parametric score function $s_{\boldsymbol{\theta}}$ and $\gamma_{\boldsymbol{\theta}}$, *iii)* the computation of the ELBO and finally *iv)* the expression for the backward process.

The forward process defined in Equation (20) has expression:

$$\mathrm{d}X_t\,[k/N] = \mathfrak{I}\left(b^l \mathfrak{F}(X_t[i/N])^l\right)^k \mathrm{d}t + \mathrm{d}W_t\,[k/N], \quad k = 1, \ldots, |Z|, \tag{30}$$

where $\mathrm{d}W_t\,[k/N] \simeq \mathfrak{F}(\mathrm{d}W_t^i)^k$. Simple calculations show that $X_t\,[k/N]$ is equivalent in distribution to

$$X_t\,[k/N] = \mathfrak{I}\left(\exp(b^l t)\mathfrak{F}(X_0[i/N])^l + \sqrt{s^l}\epsilon^l\right)^k, \tag{31}$$

where $s^l = \langle \mathcal{S}(t), e^l \rangle = r^l \frac{\exp(2b^l t) - 1}{2b^l}$ and $\epsilon^l \sim \mathcal{N}(0,1)$, allowing simulation of the forward process in a single step. The parametric score function can be approximated as:

$$s_{\boldsymbol{\theta}}\left(\sum_i X_t\,[i/N]\,\xi^i, t\right)[i/N] = \tag{32}$$

$$-\mathfrak{I}\left(\frac{\mathfrak{F}\left(X_t\,[i/N]\right)^k - \exp(b^k t)\mathfrak{F}\left(n(g(X_t\,[l/N]), t, \boldsymbol{\theta})\,[l/N]\right)}{s^k}\right)^i.$$

Similarly:

$$\tilde{\gamma}_{\boldsymbol{\theta}}\left(\sum_i X_t\,[i/N]\,\xi^i, \sum_i X_0\,[i/N]\,\xi^i, t\right)[i/N] = \tag{33}$$

$$-\mathfrak{I}\left(\frac{\exp(b^k t)}{s^k}\left(\mathfrak{F}\left(n(g(X_t\,[l/N]), t, \boldsymbol{\theta})\,[l/N] - X_0[l/N])^k\right)\right)^i.$$

---

[3]We stress that although we should consider a real Hilbert space, we select the complex exponential to avoid cluttering the notation. It is possible to select $\{\cos(2\pi p), \sin(2\pi p), \cos(2\pi 2p), \sin(2\pi 2p), \ldots\}$ as a basis, and redoing the calculations in this Section we can obtain a functionally equivalent scheme as the one with the real basis.

Combining Equation (31) and Equation (33) we can fully characterize the training objective defined in Equation (19). Then, it is possible to optimize the value of the parameters $\boldsymbol{\theta}$ with any gradient-based optimizer.

Finally, the backward process approximation is expressed as:

$$
\mathrm{d}\hat{X}_t\,[k/N] = -\Im\left(b^l \mathfrak{F}(\hat{X}_t[i/N])^l\right)^k + \Im\left(r^l \mathfrak{F}\left(s_{\boldsymbol{\theta}}(\sum_i \hat{X}_t\,[i/N]\,\xi^i, T-t)\,[i/N]\right)^l\right)\mathrm{d}t + \mathrm{d}W_t\,[k/N]
$$

(34)

$$
k = 1,\ldots,|Z|,
$$

from which new samples can be generated.

### D.1 Proofs

We start by proving Equation (30). Starting from the drift term of Equation (20), we have the following chain of equalities:

$$
\left\langle \mathcal{A}\sum_{i=0}^{N-1} X_t[i/N]\xi^i, \zeta^k \right\rangle = \left\langle \sum_{i=0}^{N-1} X_t[i/N]\mathcal{A}\frac{1}{N}\sum_{l=0}^{N-1} e^l \exp\left(-j2\pi l\frac{i}{N}\right), \zeta^k \right\rangle =
$$

$$
\left\langle \sum_{i=0}^{N-1} X_t[i/N]\frac{1}{N}\sum_{l=0}^{N-1} b^l e^l \exp\left(-j2\pi l\frac{i}{N}\right), \zeta^k \right\rangle =
$$

$$
\sum_{i=0}^{N-1} X_t[i/N]\frac{1}{N}\sum_{l=0}^{N-1} b^l \exp(j2\pi lk/N)\exp(-j2\pi li/N) =
$$

$$
\sum_{l=0}^{N-1} b^l \exp(j2\pi lk/N)\mathfrak{F}(X_t[i/N])^l =
$$

$$
\Im\left(b^l \mathfrak{F}(X_t[i/N])^l\right)^i.
$$

The noise term $\mathrm{d}W_t\,[k/N]$ is approximated as:

$$
\mathrm{d}W_t\,[k/N] = \langle \mathrm{d}W_t, \zeta^k \rangle = \langle \sum_{i=0}^{\infty} \mathrm{d}W_t^i e^i, \zeta^k \rangle = \sum_{i=0}^{\infty} \mathrm{d}W_t^i \exp\left(j2\pi i\frac{k}{N}\right) \simeq \mathfrak{F}(\mathrm{d}W_t^i)^k,
$$

where we are truncating the sum. The score term has expression:

$$
s_{\boldsymbol{\theta}}(\sum_i X_t\,[i/N]\,\xi^i, t) = -(\mathcal{S}(t))^{-1}\left(\sum_i X_t\,[i/N]\,\xi^i - \exp(t\mathcal{A})n(g(X_t\,[i/N]), t, \boldsymbol{\theta})\right) =
$$

$$
-\sum_k \frac{\overbrace{\sum_i X_t\,[i/N]\,\langle \xi^i, (e^k)^\dagger \rangle}^{=\mathfrak{F}(X_t[i/N])\overset{\text{def}}{=}C_t^k} - \exp(b^k t)\,\langle n(g(X_t\,[i/N]), t, \boldsymbol{\theta}), (e^k)^\dagger \rangle}{s^k} e^k =
$$

$$
-\sum_k \frac{C_t^k - \exp(b^k t)\,\langle n(g(X_t\,[i/N]), t, \boldsymbol{\theta}), \exp(-j2\pi kp) \rangle}{s^k} e^k \simeq
$$

$$
-\sum_k \frac{C_t^k - \exp(b^k t)\,\left(N^{-1}\sum_r n(g(X_t\,[i/N]), t, \boldsymbol{\theta})\left[\frac{r}{N}\right], \exp\left(-j2\pi k\frac{r}{N}\right)\right)}{s^k} e^k,
$$

where the approximation is due to the substitution of explicit scalar product with the discretized version trough $\mathfrak{F}$. When evaluated on the grid of interest:

$$
s_{\boldsymbol{\theta}}\left(\sum_i X_t\,[i/N]\,\xi^i, t\right)[i/N] =
$$

$$
-\sum_k \frac{\left(C_t^k - \exp\left(b^k t\right)\left(N^{-1}\sum_r n(g(X_t\,[i/N]), t, \boldsymbol{\theta})\left[\frac{r}{N}\right], \exp\left(-j2\pi k\frac{r}{N}\right)\right)\right)}{s^k}\exp(j2\pi ki/N) =
$$

$$
-\mathfrak{I}\left(\frac{\mathfrak{F}\left(X_t\,[i/N]\right) - \exp\left(b^k t\right)\mathfrak{F}\left(n(g(X_t\,[i/N]), t, \boldsymbol{\theta})\,[i/N]\right)}{s^k}\right).
$$

The value of $\tilde{\gamma}_{\boldsymbol{\theta}}$, Equation (33) and the expression of the backward process, Equation (34), are obtained similarly, considering the above results.

## E  Implementation Details and Additional Experiments

In all experiments we use the the complex Fourier basis for the Hilbert spaces, indexed by $k$. This extends to the 2-dimensional case what we described in Appendix D.1. As stated in the main paper, our practical implementation sets $f = 0$: then, we only need to specify the value for the parameters $b^k, r^k$. In our implementation we consider an extended class of SDEs that include time-varying multiplying coefficients in front of the drift and diffusion terms, as done for example in the Variance Preserving SDE originally described by Song & Ermon (2020). This can be simply interpreted as the time-rescaled version of autonomous SDEs.

### E.1  Architectural details

**INR-based score network.** In our implementation, we use the original INR architecture (Sitzmann et al., 2020). For the specific denoising task we consider in our model, we extend the input of the network architecture to include the corrupted version of the input sample and the diffusion time $t$, in addition to the spatial coordinates. We emphasize that our architectural design is simple, and does not require self-attention mechanisms (Song & Ermon, 2020). The non-linearity we use in our network is a Gabor wavelet activation function (Saragadam et al., 2023). Furthermore, we found beneficial the inclusion of skip connections.

As stated in the main paper, we consider the *modulation* approach to INRs. In particular, we implement the meta-learning scheme described by Dupont et al. (2022b); Finn et al. (2017). The outer loop is dedicated to learning the base parameters of the model, while the inner loop focuses on refining the base parameters for each input sample. In the outer loop, the optimization algorithm is AdaBelief (Zhuang et al., 2020), sweeping the learning rate over 1e-4, 1e-5, 1e-6. We found the use of a cosine warm-up schedule to be beneficial for avoiding training instabilities and convergence to sub-optimal solutions. The inner loop is implemented by using three steps of stochastic gradient descent (SGD).

**Transformer-based score network.** In our experiments with the Transformer architecture for score modeling, we employed the UViT backbone Bao et al. (2022). This backbone processes all inputs, be they temporal or noisy image patches, as tokens. Rather than utilizing UViT's default learned positional embeddings, we adapted it to integrate 2D sinusoidal positional encodings. For the noisy input images, patch embeddings transform them into a sequence of tokens. Notably, we chose a patch size of 1 to fully harness the functional properties of our framework. Time embeddings are computed based on the time and then concatenated with the image tokens.

Our chosen transformer architecture comprises 7 layers, with each layer composed of a self-attention mechanism with 8 attention heads and a feedforward layer. Furthermore, we use long skip connections between the shallower and deeper layers, as outlined by Bao et al. (2022).

For optimization during our training, we utilized the AdamWLoshchilov & Hutter (2017) algorithm with a weight decay of 0.03. We employed a cosine warm-up schedule for the learning rate, which ends at a value of 2e-4.

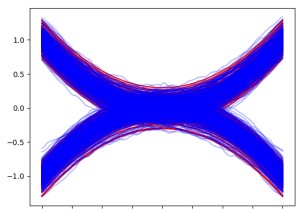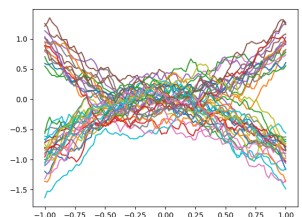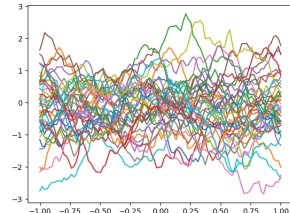

**Figure 3:** Left: real (red) and generated samples (blue). Center and Right: Samples diffused for times 0.2 and 1.0 respectively.

## E.2 Additional results

### E.2.1 A Toy example.

We here present some qualitative examples on a synthetic data-set of functions $\in L([-1, 1])$, and therefore consider the settings described in Appendix D. The *Quadratic* data is generated as in (Phillips et al., 2022), i.e. $X_0[p] = qp^2 + \epsilon$, where $\epsilon \sim \mathcal{N}(0, 0.1)$ and $q$ is a binary random variable that take values $\{-1, 1\}$ with equal probability. Concerning the design of the forward SDE, we select $b^k = \min(\sqrt{k}, 10)$ and $r^k = k^{-2}$ (thus satisfying Corollary 1). The real data is generated considering a grid of 100 equally spaced points. We can see in Figure 3 some qualitative results. On the left real (red) and generated through FDP (blue) samples show good agreement. Center and right plots depict some example of diffused samples for times 0.2 and 1.0 respectively.

### E.2.2 MNIST data-set

We evaluate our approach on a simple data-set, using MNIST $32 \times 32$ (LeCun et al., 2010). In this experiment, we compare our method against the baseline score-based diffusion models from Song et al. (2021), which we take from the official code repository `https://github.com/yang-song/score_sde`. The baseline implements the score network using a U-NET with self-attention and skip connections, as indicated by current best practices, which amounts to $O(10^8)$ parameters.

Instead, our method uses a score-network/INR implemented as a simple MLP with 8 layers and 128 neurons in each layer. The activation function is a sinusoidal non-linearity (Sitzmann et al., 2020). Our model counts $O(10^5)$ parameters. We consider an SDE with parameters $r^{k,m} = \frac{176}{k^2+m^2+2}$,[4] and $b^{k,m} = \min((k^2 + m^2 + 0.3)^{-1} + \left(\frac{r^{k,m}}{33}\right)^{\frac{1}{4}}, 3.6)$. These values have been determined empirically by observing the power spectral density of the data-set, to ensure a well-behaved Signal to Noise ratio evolution throughout the diffusion process for all frequency components.

---

[4]Strictly speaking, the sum of the series $r^{k,m}$ is not convergent. We experimented changing the decay to ensure convergence, but we observed no numerical difference with the settings we the setting we used. It is an interesting avenue for future work to study if this approximation has an impact for higher-resolution data-sets.

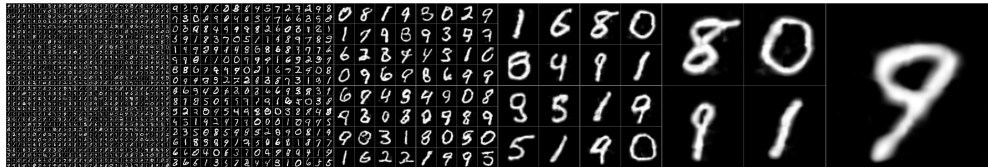

**Figure 6:** Example of super-resolution of Mnist images. From left to right: initial (training) resolution to higher resolutions.

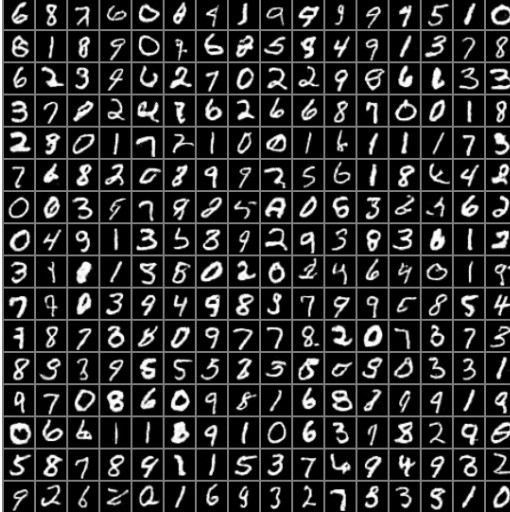

**Figure 4:** MNIST samples generated according to our proposed FDPs.

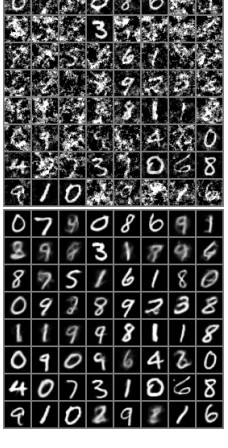

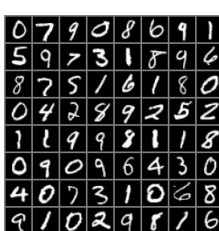

**Figure 5:** Top right: MNIST real samples. Top Left: Each sample is diffused for a given random time. Bottom: output of INR for corresponding input noisy image.

In Figure 4 we report un-curated samples generated according to our FDP. In Figure 5 we present instead various "intermediate" noisy versions of the training data, to illustrate the kind of noise we use to train the score network, and the output of the denoising INR. We also report the Fréchet Inception Distance (FID) score computed using 16k samples (lower is better). For the baseline we obtain FID=0.05, whereas for the proposed method we obtain FID=0.43. Although the FID score is in favor of the baseline, we believe that our results – obtained with a simple MLP – are very promising, as further corroborated by experiments on a more complex dataset, which we show next.

**Super resolution.** We demonstrate how an INR trained at a 32x32 resolution on MNIST can be seamlessly applied to increase the resolution of the generated data points. Given that the INR establishes a mapping between a grid and its corresponding value, we initiated the diffusion process from a grid at a 32x32 resolution. The diffusion process continued until the final step, where we used the last learned parameters to extrapolate outputs at a higher resolution. A significant advantage of using INRs is the ability to conduct the resource-intensive sampling at a lower resolution and then effortlessly transition between resolutions with just a single forward call to the model. Figure 6, shows our results at different resolutions.

### E.2.3 CELEBA data-set

For the CELEBA data-set we considered the same SDE as for the MNIST experiment. Results reported in the main paper have been obtained using a numerical integration scheme of a variant of the predictor-corrector scheme of (Song & Ermon, 2020), which we adapted to the SDEs we consider in our work. In Figure 8 and Figure 9 we report additional un-curated samples obtained with the INR and Transformer respectively. We proceed to describe further experiments in the following section.

**Conditional generation.** In the following, we consider three use-cases for conditional generation: in-painting, de-blurring, and colorization, which we describe next. All these additional experiments

were completed using the same architecture and configuration of the unconditional generation described above.

**In-painting.** We perform in-painting experiments by adopting the same approach described by Song & Ermon (2020), and report results in Figure 10. Original images (left-column of Figure 10) are masked (center-column of Figure 10), where we set the value corresponding to the missing pixels to 0. The right column of Figure 10 shows the results of the in-painting scheme where, qualitatively, it is possible to observe that the conditional generation is able to fill the missing portion of the images while maintaining good semantic coherence.

**De-blurring.** Our FDPs are naturally suited for the de-blurring use-case, as shown in Figure 12. In this experiment, we take the original images (left column of Figure 12) and filter them with a low pass filter (center column of Figure 12). The de-blurring scheme is implemented as the in-painting approach described by Song & Ermon (2020), where the only difference is that the masking at each update is applied in the frequency domain. The right column of Figure 12 shows that our technique gracefully recovers missing details and is capable of producing high quality images conditioned on the distorted inputs.

**Colorization.** In this use-case, we adapt the approach from (Song & Ermon, 2020) to our setting. Figure 11 depicts qualitative results of the colorization experiment, confirming the flexibility of the proposed scheme.

### E.2.4 SPOKEN DIGIT data-set

To demonstrate the versatility of our framework, we conducted preliminary experiments on an audio dataset, specifically the Spoken Digit Dataset. This dataset comprises recordings of spoken digits, stored as wav files at an 8kHz sampling rate, with each recording trimmed to minimize silence at the beginning and end. The dataset features five speakers who have contributed to a total of 2,500 recordings, providing 50 recordings of each digit per speaker. The dataset is publicly available on the TensorFlow Dataset Catalog. As preprocessing, each sample was either padded with zeros or truncated to a maximum duration of one second. Subsequently, the data was normalized using the effects.normalize function from the pydub library, and each sample was scaled to a range of [-1, +1] by dividing it by the dataset's maximum intensity. For the audio experiments, we fed the raw audio data directly into the transformer model, without converting it to log mel spectrograms, which is a common practice in audio processing tasks. The transformer model was configured with a patch size of 2, an embedding size of 512, 13 layers, and 8 heads. We employed the AdamW optimizer with a weight decay of 0.03 and a cosine warmup schedule that decays at a value of 1e-5. The preliminary examination of the audio generated by our model reveals its ability to effectively generate spoken digits. Upon listening to the model's generated samples, we were able to recognize the digits accurately, showcasing the model's potential in audio generation tasks. Figure 7 provides a comparative analysis of the waveforms generated by our model against real examples from the dataset.

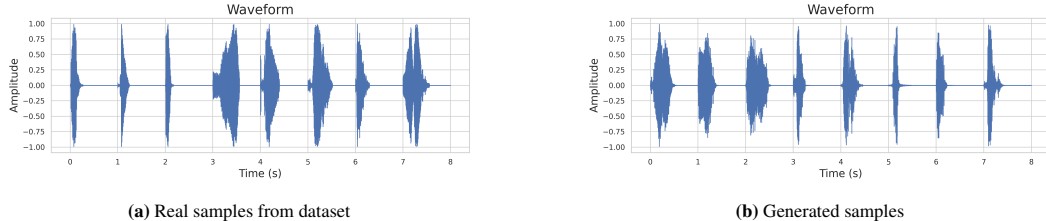

(a) Real samples from dataset     (b) Generated samples

**Figure 7:** Comparison of real and generated waveforms.

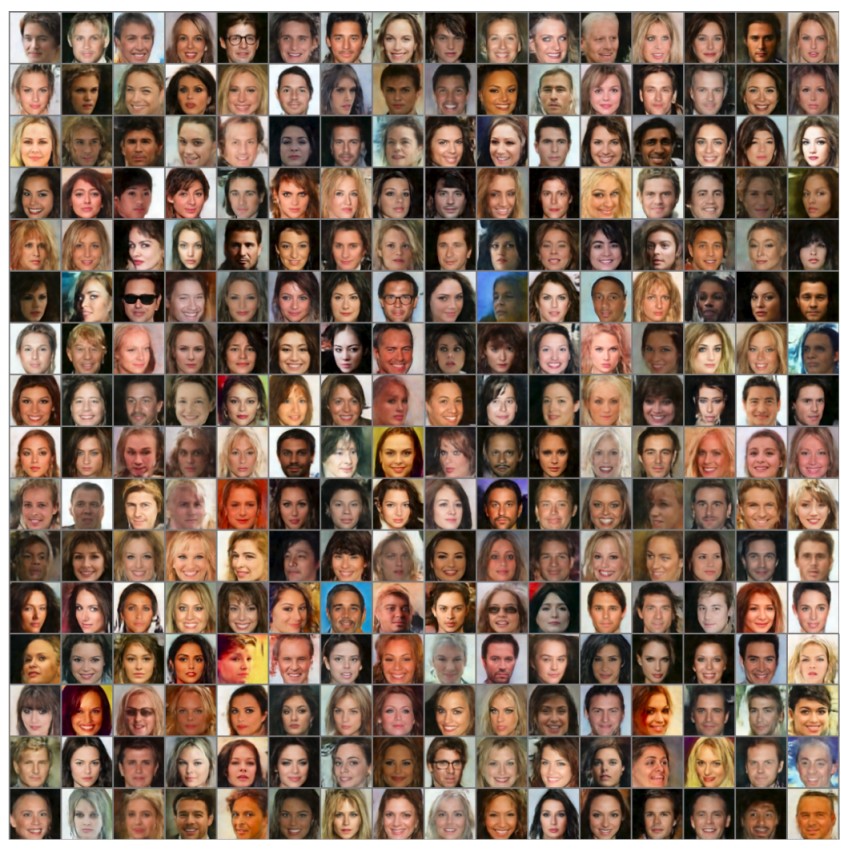

**Figure 8:** Uncurated CELEBA samples generated by the INR.

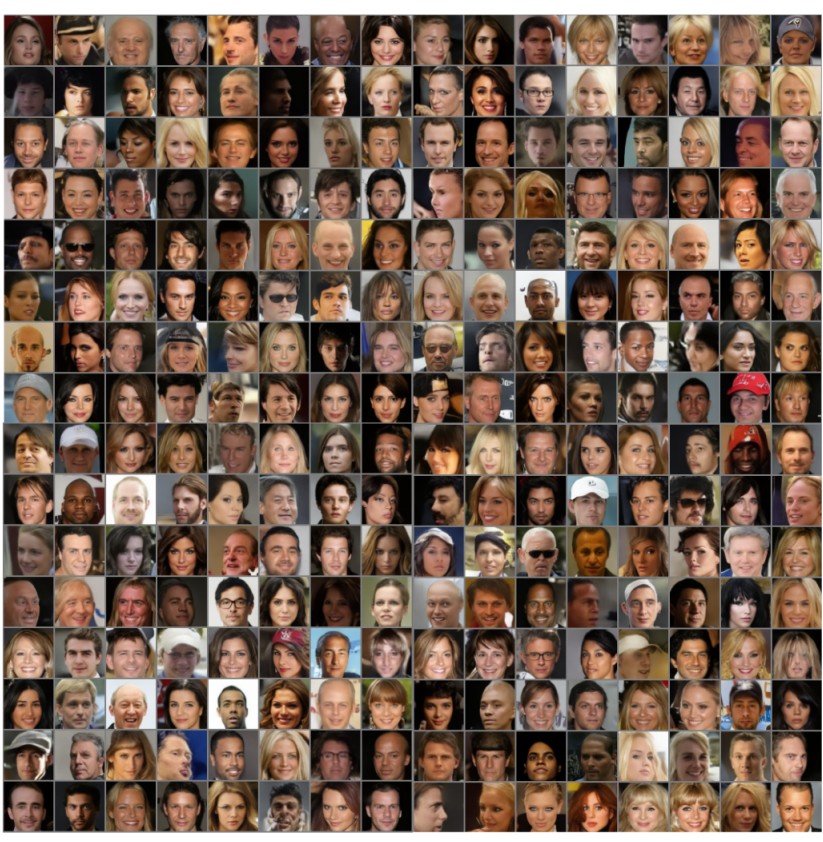

**Figure 9:** Uncurated CELEBA samples generated by the Transformer.

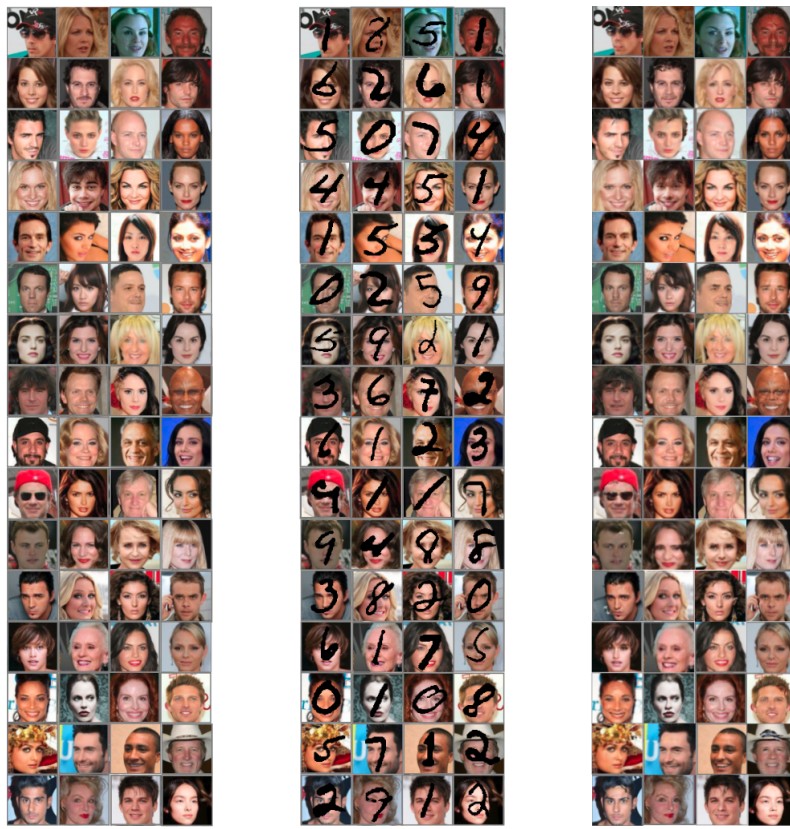

**Figure 10:** In-painting experiment using INR. Left: real samples, Center: Masked samples, Right: Reconstructed samples.

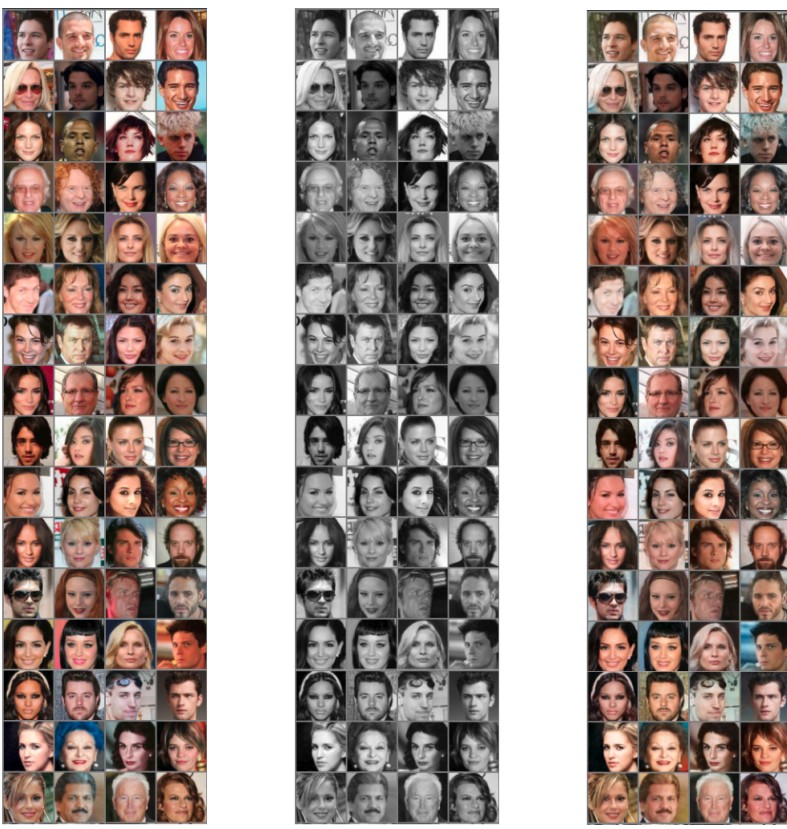

**Figure 11:** Colorization experiment using INR. Left: real samples, Center: Gray-scale samples, Right: Reconstructed samples.

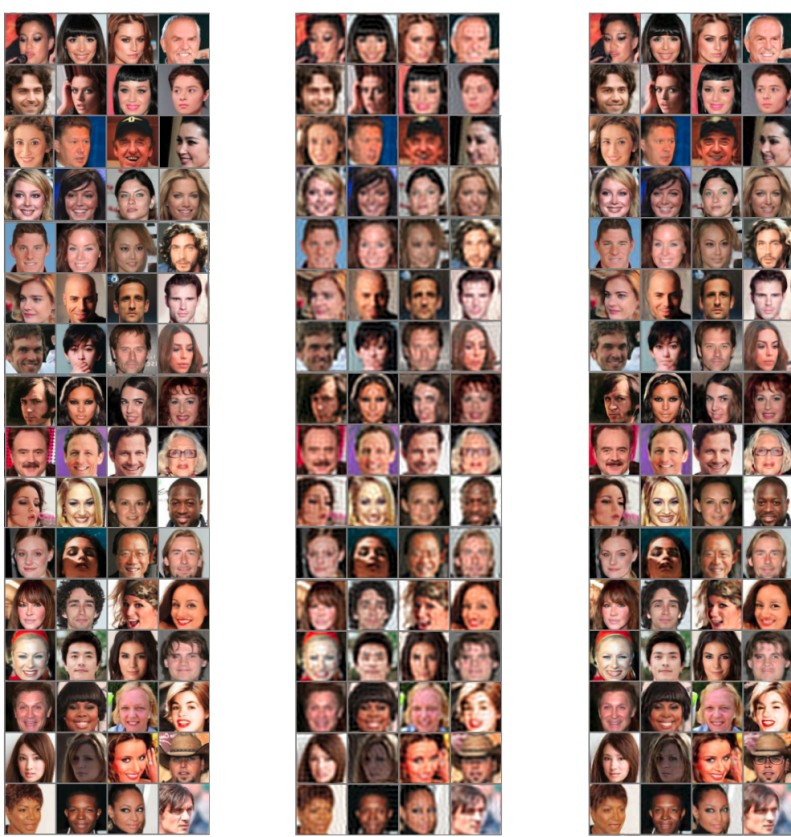

**Figure 12:** De-blurring experiment using INR. Left: real samples, Center: blurred samples, Right: Reconstructed samples.

