# OpenReview forum: "Continuous-Time Functional Diffusion Processes"
_NeurIPS.cc/2023/Conference — NeurIPS 2023 poster_

### Official Review · Reviewer_vNhS · 2023-07-04

**Soundness:** 3 good
**Presentation:** 3 good
**Contribution:** 2 fair
**Rating:** 6
**Confidence:** 4

**Summary:**

In this work the authors propose an extension to the diffusion models framework for function spaces.
In particular, given a Hilbert space $H$, they introduce a noising process based on a Wiener process on $H$. Depending on some assumptions on the type of noise, they then show the existence of the time-reversal process. Then they derive an ELBO via an extension of the Girsanov theorem.
Later, akin to the the classical Shannon-Nyquist sampling theorem, they show that for a subclass of functions and grids for which the function is uniquely determined by its values on this grid.
Eventually, the process is discretised at the evaluation points, yielding finite dimensional noising and denoising processes.
They apply their model to model the CelebA dataset, and show that with relatively few parameters their are able to obtain low FID.

**Strengths:**

- I found it quite easy to follow the submission, especially Section 2 which I think is well fleshed out.
- The proposed methodology is rigorous and sound.
- Theorem 2 is nice and potentially useful as it gives sufficient conditions to recover the original function.

**Weaknesses:**

- Perhaps the main weakness is the limited experimental setups and lack of ablation studies. As such it is hard to really understand the practical benefits of this specific approach.
- In particular, one advantage of working with functional data is to be able to handle data at discretised at arbitrary resolutions or on varying and irregular grid, yet this work practically focuses on the CelebA dataset. It is interesting to see the parameter efficiency by representing as the data as function, but indeed there is no hope to beat method that assume this fixed grid setting and can rely on specialised architectures (e.g. U-net). It may be of interest to tackle tasks such as upscaling, or data with missing values, irregular time-series etc.

**Questions:**

- line 98: Why does $R$ has to be diagonal?
- line 105: Is it correct to interpret the 'trace-class' as the following: we have $\mathrm{Tr}(R) = \sum_i \lambda_i = Var(W_t)$ that is the trace of the operator is given by the arithmetic series of its eigenvalue which itself is the variance of the $R-Wiener$ process, so this can be seen as a finite variance constraint? Which seems pretty natural.
- Equation 3: How can we define a density $\rho_t$ since there is no 'Lebesgue-like' (translation invariant) measure on infinite dimensional Hilbert space? Is it w.r.t. a Gaussian measure?
- Table 1: It is nice to see that FDP can be parameter efficient. Is the reason for not trying larger architecture computational? If so what's the bottleneck? Would $\inf$ - DIFF with 1M parameter be performing as well?
- line 333: The MLP architecture seems very deep, have you tried something perhaps a bit shallower yet wider (e.g. 512 neurons)?
- Section 6: What noise $R$ was used for this experiment? From a Karhunen-Loeve perspective would make sense to use the correlation of the data process see [Angus et al 2022].


Spectral Diffusion Processes, Phillips, Angus and Seror, Thomas and Hutchinson, Michael and De Bortoli, Valentin and Doucet, Arnaud and Mathieu, Emile, 2022

**Limitations:**

- Something that has not been explored, yet can be left , is conditional sampling which I believe is often of interest e.g. looking at the neural processes literature.

---

> ### Author Rebuttal · Authors · 2023-08-08
>
> *Perhaps the main weakness is the limited experimental setups and lack of ablation studies. As such it is hard to really understand the practical benefits of this specific approach.
> In particular, one advantage of working with functional data is to be able to handle data at discretised at arbitrary resolutions or on varying and irregular grid, yet this work practically focuses on the CelebA dataset. It is interesting to see the parameter efficiency by representing as the data as function, but indeed there is no hope to beat method that assume this fixed grid setting and can rely on specialised architectures (e.g. U-net). It may be of interest to tackle tasks such as upscaling, or data with missing values, irregular time-series etc.*
>
> We do agree with the reviewer that additional experimental validation could strengthen the work. Please refer to the shared answer, as well as the additional pdf, where extra experiments are provided.
>
> *Questions: line 98... has to be diagonal?*
>
> We do apologize for the imprecision: to define a “diagonal” operator one must specify a basis. There is no requirement on the operator R to be diagonal, it is just a choice that simplifies the design of the SDEs  coefficients. The only restriction we consider when dealing with Hilbert spaces of integrable functions is to have trace-class noise, which, as correctly guessed by the reviewer, just implies a finite variance condition
>
> *line 105: Is it correct to interpret the 'trace-class' as the following... Which seems pretty natural.*
>
> This is indeed correct (see also point above) and is an important requirement which if not satisfied can hinder existence of valid diffusion processes, a matter not explored by some of our competitors. We will clarify in the text of the revised manuscript.
>
> *Equation 3: How can we define a density since there is no 'Lebesgue-like' (translation invariant) measure on infinite dimensional Hilbert space? Is it w.r.t. a Gaussian measure?*
>
> We apologize for the technical confusion. It is indeed true that on infinite dimensional spaces there is not an equivalent of the Lebesgue measure, and this is the reason why e.g. [Lim2023] it is only possible to define the ratio w.r.t. some reference measure (like the Gaussian one).
> In our case, our compact notation indicates the conditional density $\rho^{(d)}_t(x^i | x^{ j\neq i})$. This on the contrary is a single dimensional density which exists (it is the same object considered in [Pidstrigach2023]). In the submitted version of the manuscript (line 110-111) we explicit this “To avoid cluttering the notation, we shorten ...” We do understand however that this is an important technical point, and we will expand the notation in the final version of the manuscript
>
> *Table 1: It is nice to see that FDP can be parameter efficient. Is the reason for not trying larger architecture computational? If so what's the bottleneck? Would inf- DIFF with 1M parameter be performing as well?*
>
> We explored deeper MLP architectures and found no substantial benefits. Concerning Infty-Diff, as stressed in the main paper, the considered architectures are complex combinations of functional and non functional blocks (like convolutional UNet transformers). Consequently, it is not straightforward to reduce the number of parameters without the need to completely re-design the whole architecture.
>
> *line 333: The MLP architecture seems very deep, have you tried something perhaps a bit shallower yet wider (e.g. 512 neurons)?*
>
> Yes, and we did not obtain better results. See also the shared answer above.
>
> *Section 6: What noise
>  was used for this experiment? From a Karhunen-Loeve perspective would make sense to use the correlation of the data process see [Angus et al 2022].*
>
>  We thank for the interesting question. All details about the actual values of b and r are reported in the supplementary material. Our procedure for the selection of coefficients of b and r is indeed intimately related to the observation of the power spectral density of the data. The design should satisfy the following requirements: Signal to Noise ratio (on a frequency by frequency basis) should decrease gracefully up to a low snr regime (which corresponds, roughly speaking, to the uninformative gaussian steady state) and the selection of coefficients must satisfy the requirements of corollary 1. We think it will be an interesting complement to the paper to express the connection between our methodology and the one described in [Angus2022].
>
>
>
> Spectral Diffusion Processes, Phillips, Angus and Seror, Thomas and Hutchinson, Michael and De Bortoli, Valentin and Doucet, Arnaud and Mathieu, Emile, 2022
>
> *Limitations:
> Something that has not been explored, yet can be left , is conditional sampling which I believe is often of interest e.g. looking at the neural processes literature.*
>
>  In the submitted version of the manuscript we included some conditional sampling experiments like deblurring, inpainting and colorization. We moreover include in this rebuttal some super-resolution experiments conditioned on lower resolution images. Class-conditional experiments, possible with the proposed method with minor modification to the considered architectures, are an interesting proposal for future works.

---

> > ### Comment · Reviewer_vNhS · 2023-08-18
> > **response**
> >
> > Thanks for the clarifications!
> >
> > > We explored deeper MLP architectures and found no substantial benefits
> >
> > Would you have any idea why? Can the architecture of Infty-Diff be used here?

---

> > > ### Author Response · Authors · 2023-08-19
> > >
> > > *Would you have any idea why?*
> > >
> > > We observed that deeper INR architectures suffer from meta-learning instabilities. On the other hand, Transformer based architectures are more stable when increasing the depth of the models, and show better performance.
> > >
> > > *Can the architecture of Infty-Diff be used here?*
> > >
> > > While it is certainly possible to consider a more specialized architecture, in the current work we focus on simple and modality agnostic architectures.
> > >
> > > We consider a complete analysis of the different existing trade-offs an interesting topic for future works.

---

### Official Review · Reviewer_XJB7 · 2023-07-04

**Soundness:** 4 excellent
**Presentation:** 3 good
**Contribution:** 4 excellent
**Rating:** 7
**Confidence:** 4

**Summary:**

This work proposes *functional diffusion processes (FDPs)*, which generalizes the SDE-based continuous-time framework for diffusion models for data living in Hilbert spaces. The work builds on the theory of infinite-dimensional SDEs and their time reversals, as well as an application of the infinite-dimensional Girsanov theorem to derive an ELBO objective. A practical implementation, via discretization and implicit neural representations, is proposed and evaluated on the Celeb-A dataset.

**Strengths:**

- The paper is generally clear and well-written throughout.
- This work adds to the growing literature on function-space diffusion models (a quite active area) and will likely be of significant interest to the machine learning community. The main theoretical portion of this work places continuous-time, SDE-based functional diffusion models on solid theoretical grounds.
- All theoretical claims throughout the paper are precisely stated and rigorously justified. To the best of my knowledge, the proofs of the claims (contained in the appendix) are correct.

**Weaknesses:**

- The experiments (section 6) are the weakest part of the paper.
	- The proposed methodology obtains significantly worse FID scores than standard (Euclidean) diffusion models. However, note that this is achieved with far fewer parameters.
	- It was also somewhat unclear how the "complexity" column in Table 1 was derived, as this seems like a highly subjective measurement
	- Also in Table 1, it was unclear why FID scores were being compared against FID-CLIP scores (for the model of Bond-Taylor & Willcocks, 2023).
	- It was unclear if the parameter counts in Table 1 also included the parameters for the INRs, or merely the parameters for the score network.
- The relationship between this work and the various other concurrent function-space diffusion model works could be better clarified.


#### Minor comments
- The notation on line 169 was somewhat unclear to me on a first read, particularly what $\hat{\mathbb{Q}}_0, \hat{\mathbb{P}}_0$ represent
- There may be a small typo on line 169: should it read $d \hat{\mathbb{P}}^{\chi_T} = d \hat{\mathbb{P}}^{\rho_T} \frac{d \chi_T}{d \rho_T}$?

**Questions:**

- The work restricts its attention to classes of functions which can be reconstructed exactly via finite sets of samples (Sect. 3). If my understanding is correct, the observation points may vary across functions. Does the proposed practical implementation allow for this? If so, how would you expect performance of the model to change if one were to e.g. train on one discretization and sample on another, such as an image super-resolution task?

**Limitations:**

The limitations of the work are adequately addressed in the paper.

---

> ### Author Rebuttal · Authors · 2023-08-08
>
> *The experiments (section 6) are the weakest part of the paper.
> The proposed methodology obtains significantly worse FID scores than standard (Euclidean) diffusion models. However, note that this is achieved with far fewer parameters.*
>
> Indeed, even if FID scores obtained with FDPs are not SOTA, the variant of the score network implemented as an INR achieves remarkably good image quality with several orders of magnitude fewer parameters than SOTA methods, which can be important in some application scenarios (e.g. limited resource devices). Moreover, our new set of experimental results obtained with Transformer-based score nets substantially improve FID scores, at the expense of a larger number of parameters.
> Note also that the FID metric is not sufficiently reliable, despite being widely adopted in the literature . Indeed, it is prone to being heavily influenced by small perturbations that are imperceptible to humans [Gaurav]. We include some examples of non curated samples obtained with a different functional architecture (Transformers), showing that the quality of generated images is higher than what the simple FID analysis would suggest.
>
> *It was also somewhat unclear how the "complexity" column in Table 1 was derived, as this seems like a highly subjective measurement*
>
> We do agree that a single metric about complexity could be interpreted as subjective. We will include in the camera ready version of the paper details about the architectures of all the considered competitors instead of simply using general adjectives, because this is what we meant by “complexity”. For example, the work discussed in “Bond-Taylor & Willcocks, 2023” requires multiple blocks, like classical UNeT architectures.
>
> *Also in Table 1, it was unclear why FID scores were being compared against FID-CLIP scores (for the model of Bond-Taylor & Willcocks, 2023)*
>
> Classical FID score is not available for these competitors, so we just wanted to point that out for the reader.
>
> *It was unclear if the parameter counts in Table 1 also included the parameters for the INRs, or merely the parameters for the score network.*
>
> We clarified this in the shared answer: the implicit neural representation network IS the score network. So, this is the total number of parameters we have in our implementation. Using an INR to implement the score network is possible due to the alignment of the goal of score matching (denoising task) and the intrinsic denoising capabilities of INRs (Kim et al., 2022a)
> The relationship between this work and the various other concurrent function-space diffusion model works could be better clarified.
> We do agree, see also the answer to  reviewer GA6N. We will dedicate more space in the camera ready version of the paper to such analysis.
>
> *Minor comments
> The notation on line 169 was somewhat unclear to me on a first read*
>
> We apologize for the confusion, we added some extra english wording in the paper to clarify this
>
> *There may be a small typo on line 169*
>
> Yes, thank you for spotting the error.
>
> *The work restricts its attention to classes of functions which can be reconstructed exactly via finite sets of samples (Sect. 3). If my understanding is correct, the observation points may vary across functions. Does the proposed practical implementation allow for this? If so, how would you expect performance of the model to change if one were to e.g. train on one discretization and sample on another, such as an image super-resolution task?*
>
> As explained in the shared part of the answer, we have many new results on this theme, with a positive answer to the raised questions.
>
>
> Parmar, Gaurav, Richard Zhang, and Jun-Yan Zhu. "On aliased resizing and surprising subtleties in gan evaluation." Proceedings of the IEEE/CVF Conference on Computer Vision and Pattern Recognition. 2022.

---

> > ### Comment · Reviewer_XJB7 · 2023-08-14
> >
> > Thank your for the detailed response and clarifications.
> >
> > After reading the other reviewer's comments, I still believe that this paper has strong contributions and will be of significant interest to the NeurIPS community. My score remains a 7 (accept).

---

### Official Review · Reviewer_p4Hy · 2023-07-05

**Soundness:** 4 excellent
**Presentation:** 3 good
**Contribution:** 4 excellent
**Rating:** 7
**Confidence:** 4

**Summary:**

The paper introduces a novel continuous-time diffusion-based generative model on function space. Unlike previous works in this area, which primarily focused on discrete-time formulations, the authors concentrate on stochastic differential equations (SDE) in their approach.

The paper begins by defining the forward process, also known as the noising process, as an stochastic differential equations (SDE) on function spaces. The paper subsequently demonstrates the existence of corresponding backward processes under certain conditions, similar to the formulation of finite-dimensional score-based generative models. Notably, two constraints in the forward process are required to guarantee the existence of the reverse diffusion process, which distinguishes the current work from its finite-dimensional counterparts. Firstly, the perturbation noises should be cylindrical Wiener processes, including $R$-Wiener process with a trace-class covariance operator $R$. Secondly, an operator $\mathcal{A}$ can be present in the drift term of the forward process, which generates a strongly continuous semigroup; the semigroup will serve as a contraction map.

The paper highlights the novelty of its results by emphasizing that previous approaches have not fully explored the limits of discretization. Therefore, the provided proof offers theoretical support for the utilization of various types of integrators, unlike previous approaches.

Furthermore, the authors introduce an evidence lower bound (ELBO) on the reverse diffusion process, leveraging an extension of the Girsanov theorem. This result provides a theoretical foundation for optimal guarantees, in contrast to previous approaches that relied on heuristics to minimize score matching objectives on function space.

The paper also extends the sampling theorem, enabling the reconstruction of the original function from its evaluations at a countable set of points. This extension justifies the training of function-space models with finite observations. Consequently, the authors exploit the sampling theorem to employ implicit neural representations for the model parameterizations, unlike previous methods that predominantly relied on neural operators.

Finally, the paper demonstrates the effectiveness of the proposed method on various image generation benchmark datasets. In the experiment, they use Fourier basis functions inversely proportional to the square of the basis index for the perturbation noise.

**Strengths:**

I believe that the paper's theoretical results will make significant contributions to the machine learning community for several reasons.

First, the existence of the reverse process ensures the optimality of common model parameterizations (Markovian), specifically those that model only the drift term of the reverse process, including score modeling.

Secondly, the SDE formulation sheds light on the utilization of various integration methods for sample generation, such as DDIM or exponential integrators.

Furthermore, the derivation of the evidence lower bound (ELBO) using Girsanov results in Equations 12 and 13. In these equations, the covariance operator $R$ is presented in the norm of the $R^{1/2}(\mathcal{H})$ space. The inclusion of this additional $R$ helps address the problem encountered when defining the Kullback-Leibler (KL) divergence for previous generative models based on discrete-time diffusion in function space.

Lastly, the sampling theorem introduces intriguing open questions regarding the parameterizations of score-based generative models in function space. This opens up the possibility of employing new network architectures beyond neural operators.

**Weaknesses:**

Most of the content in the paper is good. However, some improvements are needed in the experiment.

In particular, the characteristic of the function-spaced model is its resolution invariance. However, the analysis of this aspect is missing in the paper's results. I believe that some analysis regarding the resolution-invariant property of the learned models should be included, even if it's just a small amount. In such analysis, I don't think it is necessary to strictly rely on an implicit neural representation.

**Questions:**

Unlike neural operators, implicit neural implementation is powerful, but achieving discretization invariance is challenging. How do you think is the best way to approach this?

**Limitations:**

Please refer to the comments provided in the weaknesses section.

---

> ### Author Rebuttal · Authors · 2023-08-08
>
> *Weaknesses:
> Most of the content in the paper is good. However, some improvements are needed in the experiment.
> In particular, the characteristic of the function-spaced model is its resolution invariance. However, the analysis of this aspect is missing in the paper's results. I believe that some analysis regarding the resolution-invariant property of the learned models should be included, even if it's just a small amount. In such analysis, I don't think it is necessary to strictly rely on an implicit neural representation.*
>
> Please refer to the common remark above.
>
> *Questions:
> Unlike neural operators, implicit neural implementation is powerful, but achieving discretization invariance is challenging. How do you think is the best way to approach this?*
>
> In principle, INRs and meta learning are — by design — resolution invariant. Indeed, the modulations are obtained by minimizing the average squared distance between the reconstructed and original image. This operation is by construction agnostic to the resolution or the spacing of the grid at which points are evaluated. We include in the rebuttal new experiments showcasing how the INRs can be used to produce higher resolution images.
> In practice, one can expect the meta learning procedure to present instabilities when tested at resolutions different from the training ones, as the various numerical approximation errors can cumulate and push the meta learning procedure into instability-prone regions.

---

### Official Review · Reviewer_jQYa · 2023-07-06

**Soundness:** 3 good
**Presentation:** 3 good
**Contribution:** 3 good
**Rating:** 6
**Confidence:** 1

**Summary:**

The authors propose a model approach called Functional Diffusion Processes (FDPs) that generalizes score-based diffusion model to infinite-dimensional function space. The authors derive the reverse time dynamics and sampling theorems to find a subset of functions on a countable set of samples without losing information. The authors demonstrate their approach on a multi-layered perception.


**Strengths:**

1. The theoretical parts of the paper appears to be well structured. However, the paper itself does not appear to be self contained and has many references to the appendices to understand. For example, the paper makes many references to the assumptions being made, but they are never mentioned in the main paper and are only mentioned in the appendix.


**Weaknesses:**

2. The authors claim their approach is to create a new breed of generative models in function space, which do not require specialized network architectures, and that can work with any kind of continuous data. This itself is an interesting idea. However, after reading the introduction and looking at the experimental results. I cannot see what applications this may benefit and how much of an improvement it has to have a purely functional domain. I believe this could be better motivated.
3. This may be because I didn't understand the experiment in Section 6 and does not appear to be self contained. I can understand that there is a page limitation, however, I believe that the experiment section is a bit compressed and difficult to understand. Having said that, the authors do have additional experiments in the appendix which appears to be much better structured.

**Questions:**

The theoretical parts of the paper is quite dense and I cannot say I have fully understood the paper to come up with any meaningful questions.

---

> ### Author Rebuttal · Authors · 2023-08-08
>
> *The theoretical parts of the paper appears to be well structured. However, the paper itself does not appear to be self contained and has many references to the appendices to understand. For example, the paper makes many references to the assumptions being made, but they are never mentioned in the main paper and are only mentioned in the appendix.*
>
> Thank you for raising your concern about our paper being self-contained. We understand the importance of making the paper accessible to readers.
> While we aimed to maintain a concise presentation within the given page limit, we acknowledge that striking a balance between brevity and comprehensiveness is challenging.  To improve the readability of our paper, we plan to incorporate a summary of the key assumptions in the main paper. This will provide readers with a more coherent understanding of the underlying concepts without relying heavily on appendices.
>
> *The authors claim their approach is to create a new breed of generative models in function space, which do not require specialized network architectures, and that can work with any kind of continuous data. This itself is an interesting idea. However, after reading the introduction and looking at the experimental results. I cannot see what applications this may benefit and how much of an improvement it has to have a purely functional domain. I believe this could be better motivated.*
>
> The motivations for adopting a functional perspective are multiple. First, we claim that working in a functional domain allows greater parameter efficiency and simpler architectural design, as already shown in the submitted version of the manuscript. Moreover, the approach allows using the same architectures for different modalities. We performed some preliminary experiments on audio datasets, which we will include in the camera ready, using the same exact architectures.
> Finally, the functional approach allows the design (and use) of architectures which are dimension agnostic. This claim is supported in the rebuttal by the new set of experiments concerning super-resolution tasks.
>
> *This may be because I didn't understand the experiment in Section 6 and does not appear to be self contained. I can understand that there is a page limitation, however, I believe that the experiment section is a bit compressed and difficult to understand. Having said that, the authors do have additional experiments in the appendix which appears to be much better structured.*
>
> While it is true that we left out the experimental details (like selection of parameters b and r) for the appendix, we clearly stated the considered datasets, the competitors and the metrics on which the different methods were compared. We are convinced that the information in the main paper and the appendix to be sufficient to understand our experiments. Moreover, we will release, if the paper is accepted, the full source code of the software implementation of our work, with instructions to reproduce the experiments we did, and additional details for practitioners who would be interested in using our approach.

---

### Official Review · Reviewer_GA6N · 2023-07-06

**Soundness:** 3 good
**Presentation:** 3 good
**Contribution:** 3 good
**Rating:** 6
**Confidence:** 3

**Summary:**

The authors introduce a framework for infinite dimensional diffusion models in Hilbert spaces, including deriving a reverse process and novel ELBO loss objective. The authors introduce a novel score model network architecture.

**Strengths:**

- This is a timely paper with a lot of interest in the community. There are many concurrent works also looking into similar topics e.g. [1] (I do not expect detailed comparison given they were only public recently).
- Although performance is not great compared to standard diffusion methods in terms of generative modelling, the results are better than Neural Processes and similar point wise methods (with the exception of [2], concurrent so cannot criticise for worse performance)
- The use of implicit optimization based encoder, g, in equation 16, is interesting. I have some concerns detailed below but nevertheless it is interesting.
- Theoretical results seem correct but have not been checked thoroughly

[1] Infinite-Dimensional Diffusion Models for Function Spaces, Pidstrigach et al, 2023
[2] Infinite resolution diffusion with subsampled mollified states., Bond-Taylor et al, 2023

**Weaknesses:**

- The implicitly defined INR encoder, g, detailed in equation 16, considers gradient descent in an inner layer. Will this not be quite slow and hence generation / sampling would also be quite slow?
- Empirical performance / FID scores are quite bad in comparison to other standard methods. Although interesting, is there an application where this method would be more useful than standard methods?
- Functional diffusions could be interesting for dimension agnostic diffusions, super resolution, irregular data. There are limited experiments looking into these other applications.
- Complexity in table 1 is quite subjective. I would argue parameter could does not matter if it can be ran on a single GPU in reasonable wall clock time. The implicit layer, g, with gradient descent for equation 16 is sequential so I imagine this slows down generation significantly.
- Calling other concurrent / prior work heuristic without details. "However, these works do not formally prove the existence of a backward process and their score approximation and score matching optimization objective is heuristic". Does not [1] prove existence in Theorem 1? (I do not expect detailed comparison given they were only public recently, but given the authors claim this conccurrent work is heuristic I would be interested to know more).

[1] Infinite-Dimensional Diffusion Models for Function Spaces, Pidstrigach, 2023

**Questions:**

- The implicitly defined INR encoder, g, detailed in equation 16, considers gradient descent in an inner layer. How many steps are taken? Is the gradient through the score network taken through these gradient descent steps? Is this not quite slow? Have the authors considered other point-wise encoders?

**Limitations:**

The authors have a brief paragraph on limitations regarding perform, more challenging experiments and comparisons to other approaches like NFO, but could be further discussed. See questions and weaknesses.

---

> ### Author Rebuttal · Authors · 2023-08-08
>
> *Weaknesses:
> The implicitly defined INR encoder, g, detailed in equation 16, considers gradient descent in an inner layer. Will this not be quite slow and hence generation / sampling would also be quite slow?*
>
> Please refer to the common remark for all reviewers, above.
>
> *Empirical performance / FID scores are quite bad in comparison to other standard methods. Although interesting, is there an application where this method would be more useful than standard methods?*
>
> Indeed, we obtain a FID score that is larger than the state of the art, but with several orders of magnitude fewer parameters! In addition, we have now a set of new results where the FID score is much lower (see Fig. 2, additional pdf). These new results have been obtained with an alternative implementation of the score network, based on a simple Transformer architecture. In general, the FID metric should be taken with a grain of salt, as it is not necessarily  representative of the true underlying quality of the generated data [Gaurav].
> Despite a rather involved mathematical formulation, the functional approach is interesting in particular for practitioners. Indeed, in principle, it allows working on any data modality using the same architecture for the score network. Standard methods, on the other hand, require careful design and fine tuning, which can be lengthy and expensive tasks. We have additional preliminary results on audio data, whereby FDPs use the same score network as for image data.
>
> *Functional diffusions could be interesting for dimension agnostic diffusions, super resolution, irregular data. There are limited experiments looking into these other applications.*
>
> Indeed, our goal in this paper is to develop in detail a new methodology: this requires a  careful, rigorous and lengthy analysis, which sacrifices space for empirical validation.
> Please refer also to the common remark above for additional comments and results.
>
> *Complexity in table 1 is quite subjective. I would argue parameter could does not matter if it can be ran on a single GPU in reasonable wall clock time. The implicit layer, g, with gradient descent for equation 16 is sequential so I imagine this slows down generation significantly.*
>
> First, we would like to clarify that by “complexity”, we refer to the (subjective) effort of a practitioner to design the score network architecture. Parameter count is listed separately from “complexity”. For example, the work InftyDiff, requires the cascade of multiple models (NFO, knn interpolators, Convolutional Unets), whereas the architectures we  consider in our work are much simpler (INRs are literally vanilla MLPs with sinusoidal activation functions, the new transformer based score network has no convolutional layers, etc…).
> As a reminder, we consider the implementation of the score network as an engineering choice: INR-based score nets are small and simple, but require meta-learning, the newly introduced Transformer-based score nets are larger, but do not require meta-learning.
>
> *Calling other concurrent / prior work heuristic without details. "However, these works do not formally prove the existence of a backward process and their score approximation and score matching optimization objective is heuristic". Does not [1] prove existence in Theorem 1? (I do not expect detailed comparison given they were only public recently, but given the authors claim this concurrent work is heuristic I would be interested to know more).*
>
> We apologize if the message was not clear. In the concurrent work [1], the existence of the backward process is proven, where the score functional is introduced. We are however, at the time of the submission of the manuscript, and to the best of our knowledge, the first one to show that the score matching objective is not only an heuristic but a sound variational bound. We achieve this thanks to the infinite dimensional generalization of the Girsanov theorem.
> As it has also been requested by another reviewer, we will expand more how our work differs from related works, clarifying such details.
> [1] Infinite-Dimensional Diffusion Models for Function Spaces, Pidstrigach, 2023
>
> *Questions:
> The implicitly defined INR encoder, g, detailed in equation 16, considers gradient descent in an inner layer. How many steps are taken? Is the gradient through the score network taken through these gradient descent steps? Is this not quite slow? Have the authors considered other point-wise encoders?*
>
> Please refer to the common remark above. In short, yes INR requires meta-learning which can be challenging, and yes, we considered different implementations of the score nets using simple Transformer architectures.
>
>
>
> Parmar, Gaurav, Richard Zhang, and Jun-Yan Zhu. "On aliased resizing and surprising subtleties in gan evaluation." Proceedings of the IEEE/CVF Conference on Computer Vision and Pattern Recognition. 2022.

---

> > ### Comment · Reviewer_GA6N · 2023-08-16
> >
> > Thank you for your response.
> >
> > Personally I would drop the complexity argument. It is highly subjective and somewhat misleading. One could argue that given Unets are so common across fields now, that it has the lowest "engineering" complexity, indeed practically there are many open source libraries for Unets used in score based generative models.
> >
> > Regarding using a transformer model and data agnostic approaches. There are a number of papers doing this now, one such paper [1] takes a related but somewhat heuristic approach using a transformer (perceiver) architecture and applied across modalities and for super-resolution tasks. This should also be compared (according to open review it was published in February, neurips guidance states that only work within 2 months of submission can be ignored as concurrent [2] ).
> >
> > In light of improved experiments and including audio experiments, I have increased my score to weak accept.
> >
> > [1] Diffusion Probabilistic fields, https://openreview.net/forum?id=ik91mY-2GN
> > [2] https://neurips.cc/Conferences/2022/PaperInformation/NeurIPS-FAQ

---

> > > ### Author Response · Authors · 2023-08-19
> > >
> > > We thank the reviewer for the increased score and the feedback!
> > >
> > > *Personally I would drop the complexity argument. It is highly subjective and somewhat misleading. One could argue that given Unets are so common across fields now, that it has the lowest "engineering" complexity, indeed practically there are many open source libraries for Unets used in score based generative models.*
> > >
> > > This is indeed a valid argument, which we will include and expand in our discussion.
> > >
> > > *Regarding using a transformer model and data agnostic approaches...*
> > >
> > > Thanks for the suggestion, we will include the suggested reference [1] in the final version of the manuscript, clarifying the differences w.r.t. our work.

---

### Author Rebuttal · Authors · 2023-08-08

We sincerely thank the reviewers for their thorough analysis of our work and for their very positive feedback on our paper.
We here provide some general comments which are common to all reviewers, and clarify the minor technical points directly in the messages to individual reviewers.

*Implicit Neural Representation: clarifications, discussion of limitations and considered alternatives*

First, we clarify that the considered INR effectively implements the score network, they are the same object (see lines 292-293)
In our experiments, we tried a variety of architectural choices for the INR (deep and narrow vs. shallow and wide architectures) and found, in accordance with literature results [Dupont2022b], that the former design choice consistently provides better results.
The reviewers correctly point out a possible drawback of the meta-learning approach for INRs, which require local adaptation steps. It is however important to stress that our design goal is to use as few inner steps as possible (we obtain our results with only 3 inner steps). This is in line with the approach described in [Dupont2022b].
Note that we view the design of the score network as an “engineering” choice. In the submitted paper, we list possible alternatives and present results with an INR, which has the clear benefit of requiring an extremely small number of parameters. In addition, we have new original results where the score network is implemented using a simple Transformer architecture (interpreted as a mapping between Hilbert spaces, as discussed in [Cao2021]). These new results, which we will add to a possible camera ready version of the paper, indicate improvements in terms of image quality (FID score=11, fig 2 in the  additional pdf) and do not require meta-learning steps, but require more parameters (O(40M)) than the INR variant. In our experiments with Transformers, we adopted the UViT backbone[Bao]. This backbone treats all inputs, whether time or noisy image patches, as tokens. Instead of using UViT's learned positional embeddings, we modified it to incorporate 2D sinusoidal positional embeddings as described in [Fan].

*Resolution invariance/ Different modalities*

A shared request among the reviewers is an experiment to illustrate the benefits of FDPs for an example task which can leverage the intrinsic dimensionality agnostic property of a functional representation of data.
We present new experimental results on the task of super resolution. We demonstrate how the same INR trained in the submitted version of the paper can be seamlessly applied to increase the resolution of the generated data points. (Refer to additional pdf fig 1). This is a practical approach that leverages the properties of INRs.
Moreover, we have preliminary results on different data modalities (audio waveforms) using the very same Transformer architectures considered for image data, grounding our claim that FDPs allow to simplify the design of the score network for a variety of application domains.

In principle, nothing prevents using FDPs on datasets where data has been collected on irregular grids. Indeed, the sampling theorem described in Section 3 does not require regularity of the grid but only that the covering is “sufficiently fine grained”. Moreover, score networks based on either INRs or Transformers have no problems in dealing with irregularly spaced data (to the best of our knowledge, NFOs instead require regular spacing). The major technical problem to address in case of irregular data, is the numerical simulation of the infinite dimensional SDEs, for which efficient integration schemes are elusive. In our future work, we plan to investigate such peculiar integration schemes.

*Bao, Fan, et al. "All are worth words: A vit backbone for diffusion models." Proceedings of the IEEE/CVF Conference on Computer Vision and Pattern Recognition. 2023.*

*Fan, Haoqi, et al. "Multiscale vision transformers." Proceedings of the IEEE/CVF international conference on computer vision. 2021.*

---

### Decision · Program_Chairs · 2023-09-21

**Decision:**

Accept (poster)

**Comment:**

The authors present a new framework for infinite-dimensional diffusion models valued in a Hilbert space. In particular, this framework encompasses the derivation of the reverse process of an infinite-dimensional diffusion, which generalizes the finite-dimensional setting. Then, the authors introduces a  Evidence Lower Bound (ELBO) loss objective to learn the drift of this reverse process to generate new samples. Finally, the paper introduces a novel score model network architecture.

The reviewers and I think that this paper is an interesting contribution to the diffusion models literature. As a result, we all agree that the paper should be accepted.

Please take the reviewers' comments into account when submitting the final version of your work.